# Native Logical and Hierarchical Representations with Subspace Embeddings

## Abstract

Traditional embeddings represent datapoints as vectors, which makes similarity easy to compute but limits how well they capture hierarchy, asymmetry and compositional reasoning. We propose a fundamentally different approach: representing concepts as learnable linear subspaces. By spanning multiple dimensions, subspaces can model broader concepts with higher-dimensional regions and nest more specific concepts within them. This geometry naturally captures generality through dimension, hierarchy through inclusion, and enables an emergent structure for logical composition, where conjunction, disjunction, and negation are mapped to linear operations. To make this paradigm trainable, we introduce a differentiable parameterization via soft projection matrices, allowing the effective dimension of each subspace to be learned end-to-end. We validate our approach on hierarchical and natural language inference benchmarks. Our method not only achieves state-of-the-art performance but also provides a more interpretable, geometrically-grounded model of entailment. Remarkably, the ability to perform logical composition with the learned concepts arises naturally from standard training objectives, without any direct supervision.

## 1 Introduction

Dense vector embeddings have become the bedrock of modern machine learning, underpinning systems from language models (LMs) (Devlin et al., 2019; Reimers & Gurevych, 2019) and vision-language models (VLMs) (Radford et al., 2021; Li et al., 2022), to retrieval augmented generation (RAG) systems (Lewis et al., 2020). By representing words, documents and images as points in high-dimensional space, these representations excel at capturing similarities in a scalable manner.

Despite their success, the efficacy of vector embeddings is limited by a geometric mismatch: the flat, symmetric structure of Euclidean space is ill-suited to the hierarchical and asymmetric nature of language and logic (Horn, 1972). Due to its symmetry, metrics like cosine similarity cannot capture directional relationships such as entailment or hyponymy; a high similarity between "dog" and "animal" fails to convey that one is a subtype of the other. Moreover, vector spaces lack native operators for logical conjunction and negation. This forces models to default to additive composition, effectively treating phrases as a bag-of-words. This explains why queries with negations often fail, with embeddings including the very concept meant for exclusion. Recent work confirms these flaws empirically, showing that even advanced models disregard logical connectives (Yuksekgonul et al., 2023; Moreira et al., 2025), requiring *ad-hoc* solutions (Weller et al., 2024; Gokhale et al., 2020; Zhang et al., 2025; Alhamoud et al., 2025). This inability to interpret nuanced instructions motivates our search for a framework that can natively represent these crucial relations.

We propose an alternative that extends Euclidean vector representations: instead of mapping a concept to a single vector, we embed it as a linear subspace of $\mathbb{R}^d$ *i.e.*, the span of a set of basis vectors. This enables an interpretable geometric understanding of conceptual properties. First, generality and specificity are captured by the subspace dimension, with higher-dimensional subspaces denoting broader concepts *e.g.*, animal vs. dog. Secondly, hierarchy is naturally modeled by subspace inclusion, where a more specific concept's subspace is contained within a more general one. Finally, logical operations are directly mapped to linear-algebraic operations: conjunction as subspace intersection, disjunction as linear sum (span), and negation as the orthogonal complement (Fig. 1).

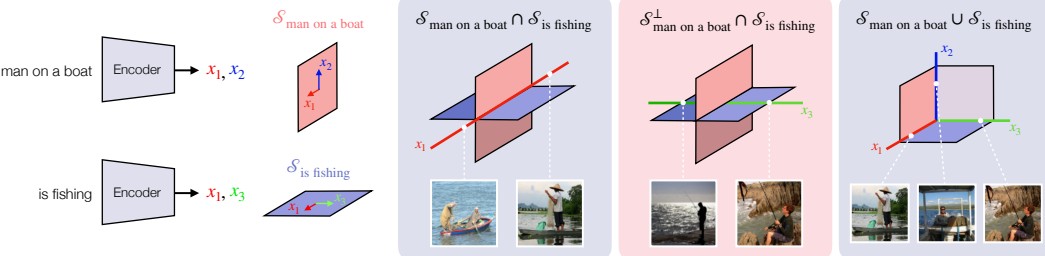

Figure 1: We embed concepts as linear subspaces of $\mathbb{R}^d$ (left). These representations enable logical operations: subspace intersections *e.g.*, "man on a boat" and "is fishing" (middle left); negation and composition *e.g.*, orthogonal complement of "of man on a boat" and "is fishing" (middle right) and linear sums of subspaces, which yield a higher variance of instances (right).

A key challenge in learning subspaces is that dimensionality, or the number of basis vectors, is discrete and thus non-differentiable. Our technical contribution overcomes this by introducing a differentiable parameterization via soft projection matrices. Instead of selecting an integer dimension, we learn a set of vectors and modulate their individual importance, allowing each subspace to add or drop basis vectors as needed during training. Crucially, this approach remains grounded in Euclidean geometry, preserving full compatibility with standard training pipelines, Euclidean metrics and loss functions. This allows for seamless integration with highly efficient, dot-product-based search libraries Douze et al. (2025); Johnson et al. (2019), ensuring our method is scalable.

Beyond quantitative performance, our approach yields representations with emergent properties that are not explicitly optimized for. Remarkably, by training solely on entailment, our model learns embeddings that are inherently amenable to logical composition, supporting operations like conjunction, disjunction, and negation of queries. We also observe a strong correlation between the learned dimensionality of a subspace and the semantic generality of the concept it represents. This provides an interpretable measure of a concept's specificity that can be leveraged for compression.

We validate our framework across standard lexical and textual entailment benchmarks. Our method sets a new state of the art on WORDNET reconstruction, shows a stronger correlation with human judgments on HyperLex, and surpasses strong bi-encoder baselines on SNLI, demonstrating robust performance and generalization.

In summary, our key contributions are:

- A novel and differentiable method for learning subspace representations of language, featuring a data-dependent dimensionality that captures semantic specificity.

- An emergent structure for logical composition over natural language. We show that fundamental logical operations, such as conjunction, disjunction, and negation, arise naturally from standard training objectives, without any explicit logical supervision.

- A demonstration that these expressive representations remain tractable for large-scale retrieval by preserving compatibility with standard, highly optimized vector search pipelines.

## 2 RELATED WORK

Most embedding methods, from Word2Vec (Mikolov et al., 2013) to multimodal models such as CLIP (Radford et al., 2021), rely on a simple idea: datapoints are represented as vectors in a high-dimensional metric space, where similarity is encoded by inner products or distances.

**Limitations of Vector Representations.** This prevalent vector-based view, while powerful for capturing co-occurrence patterns, exhibits limitations: the inner product cannot capture asymmetric relationships, such as entailment or hierarchies, without additional structural constraints or complex transformations. Recent empirical analyses have shed light on how language and vision-language encoder models represent hierarchies (Park et al., 2025; He et al., 2024) and logical constructs.

Remarkably, instead of capturing formal logical structure, vector embeddings behave akin to bag-of-words representations (Yuksekgonul et al., 2023), failing to differentiate between positive and negated concepts (Gokhale et al., 2020; Singh et al., 2024; Moreira et al., 2025; Alhamoud et al., 2025). This limitation has motivated the creation of enhanced datasets and benchmarks with explicit negations (Quantmeyer et al., 2024; Weller et al., 2024; Zhang et al., 2025).

**Hyperbolic Embeddings.** Hyperbolic embeddings (Nickel & Kiela, 2018; 2017; Ganea et al., 2018a) exploit the exponential growth of hyperbolic space to model hierarchical structures compactly and encode transitive inclusion (Bai et al., 2021). Applications include hierarchical classification (Dhall et al., 2020), logical prediction (Xiong et al., 2022), and entailment reasoning (Poppi et al., 2025). However, they require complex Riemannian optimization, lack native logical reasoning, and struggle with non-hierarchical relations (Sala et al., 2018; Moreira et al., 2024).

**Partial Order Embeddings.** Partial order embeddings (Vendrov et al., 2016; Li et al., 2017) map entities into partially ordered spaces. Variants include positive operator embeddings (Lewis, 2019), quantum logic-inspired representations (Garg et al., 2019; Srivastava et al., 2020) and probabilistic approaches such as Gaussians, Beta distributions, box lattices (Vilnis & McCallum, 2015; Athiwaratkun & Wilson, 2018; Choudhary et al., 2021; Ren & Leskovec, 2020; Vilnis et al., 2018; Li et al., 2018; Ren et al., 2020), and entailment cones (Zhang et al., 2021; Pal et al., 2025; Ganea et al., 2018b; Yu et al., 2024). While effective for entailment, these methods lack a principled logical structure, requiring *ad-hoc* losses, or relying on heuristic approximations for disjunction and negation. Our subspace embeddings overcome these limitations. While subspace inclusion models entailment, the key advantage is its emergent closure: the intersection, sum, and orthogonal complement provide principled representations for conjunction, disjunction, and negation, respectively.

## 3   SUBSPACE REPRESENTATIONS

This paper presents a paradigm shift in embeddings: rather than representing a datapoint as a single vector $\boldsymbol{x} \in \mathbb{R}^d$, we represent it as a subspace $\mathcal{S} \subseteq \mathbb{R}^d$. To illustrate, consider Fig. 1. Instead of the traditional formulation, where the concept "man on a boat" is embedded as a single direction, we map it to $\boldsymbol{x}_1$ and $\boldsymbol{x}_2$. Each vector encodes a variation of the underlying concept: $\boldsymbol{x}_1$ might represent a "man on a boat that is fishing" while $\boldsymbol{x}_2$ represents a "man on a boat that is not fishing". The concept "man on a boat" is then represented by the subspace $\mathcal{S}_{\text{man on a boat}} = \text{span}(\boldsymbol{x}_1, \boldsymbol{x}_2)$, encompassing all instances that align with either $\boldsymbol{x}_1$, $\boldsymbol{x}_2$, or any linear combination thereof, representing the space of all possible instances (Van Rijsbergen, 2004; Ganter & Wille, 2024).

Formally, we parameterize a subspace $\mathcal{S}$ as the span of $n \geq d$ learnable vectors $\boldsymbol{X} = [\boldsymbol{x}_1 \quad \ldots \quad \boldsymbol{x}_n] \in \mathbb{R}^{d \times n}$. Let the thin singular value decomposition of $\boldsymbol{X}$ be $\boldsymbol{U}\boldsymbol{\Sigma}\boldsymbol{V}^\top$, with $\boldsymbol{U} \in \mathbb{R}^{d \times r}$ and $\boldsymbol{U}^\top \boldsymbol{U} = \boldsymbol{I}_r$. Then $\boldsymbol{U}$ is an orthonormal basis for the rank-$r$ subspace $\mathcal{S}$. We can write an equivalent representation of $\mathcal{S}$ through its orthogonal projection operator,

$$\boldsymbol{P} := \boldsymbol{X}(\boldsymbol{X}^\top \boldsymbol{X})^\dagger \boldsymbol{X}^\top = \boldsymbol{U}\boldsymbol{U}^\top \in \mathbb{R}^{d \times d}, \tag{1}$$

where $\dagger$ is the pseudoinverse. This projector is symmetric ($\boldsymbol{P}^\top = \boldsymbol{P}$), idempotent ($\boldsymbol{P}^2 = \boldsymbol{P}$), and its trace reveals the rank of $\mathcal{S}$ *i.e.*,

$$\text{Tr}(\boldsymbol{P}) = \text{Tr}(\boldsymbol{U}^\top \boldsymbol{U}) = \text{Tr}(\boldsymbol{I}_r) = r. \tag{2}$$

**Subspace Similarity and Inclusion.** Cosine similarity between vectors can be generalized to subspaces $\mathcal{S}_i, \mathcal{S}_j$, with orthonormal basis $\boldsymbol{U}_i, \boldsymbol{U}_j$, respectively, via their projection operators $\boldsymbol{P}_i$ and $\boldsymbol{P}_j$,

$$\text{sim}(\boldsymbol{P}_i, \boldsymbol{P}_j) := \text{Tr}(\boldsymbol{P}_i \boldsymbol{P}_j) = \|\boldsymbol{U}_i^\top \boldsymbol{U}_j\|_F^2 = \sum_{k=1}^m \cos^2(\theta_k), \tag{3}$$

where $\{\theta_k\}_{k=1}^m$ are the *principal angles* between $\mathcal{S}_i$ and $\mathcal{S}_j$ and $m = \min\{\text{rank}(\mathcal{S}_i), \text{rank}(\mathcal{S}_j)\}$. Each $\theta_k$ is the smallest possible angle between a unit vector in $\mathcal{S}_i$ and a unit vector in $\mathcal{S}_j$, subject to orthogonality constraints on previously chosen directions. Thus, $\text{sim}(\boldsymbol{P}_i, \boldsymbol{P}_j)$ measures the total squared alignment across the $m$ most comparable directions of the two subspaces, or their degree of *overlap*. This recovers standard cosine similarity as a special case: if $\boldsymbol{P}_i$ and $\boldsymbol{P}_j$ are rank-one projectors onto unit vectors $\boldsymbol{x}_i$ and $\boldsymbol{x}_j$, then $\text{sim}(\boldsymbol{P}_i, \boldsymbol{P}_j) = (\boldsymbol{x}_i^\top \boldsymbol{x}_j)^2 = \cos^2(\angle(\boldsymbol{x}_i, \boldsymbol{x}_j))$.

An immediate consequence of Eqs. (2) and (3) is that we can quantify subspace inclusion via a normalized inclusion score (NIS) (Da Silva & Costeira, 2009):

$$\mathrm{NIS}(\boldsymbol{P}_j \mid \boldsymbol{P}_i) := \frac{\mathrm{sim}(\boldsymbol{P}_i, \boldsymbol{P}_j)}{\mathrm{Tr}(\boldsymbol{P}_i)} \in [0, 1]. \tag{4}$$

This score attains 1 if and only if subspace $i$ is contained within subspace $j$. This formulation allows for an intuitive interpretation as a Bayes-like conditional probability: the probability of an instance belonging to subspace $j$ given it belongs to $i$.

## 3.1 ALGEBRAIC STRUCTURE OF SUBSPACES

The power of subspaces lies in their algebraic structure, which natively supports interpretable operations between concepts. Using projection operators lets us map logical relations such as conjunction ($\wedge$), disjunction ($\vee$) and negation ($\neg$) into the subspace operations of intersection ($\cap$), linear sum ($+$) and orthogonal complement ($\perp$), respectively. These have tractable linear-algebraic representations, thus addressing the limitations of vector embeddings discussed in §1.

**Conjunction** ($i \wedge j$). Corresponds to the intersection of subspaces $\mathcal{S}_{i \wedge j} = \mathcal{S}_i \cap \mathcal{S}_j$. Any vector in $\mathcal{S}_{i \wedge j}$ is an element of $\mathcal{S}_i$ and $\mathcal{S}_j$. The product $\boldsymbol{P}_i \boldsymbol{P}_j$ is an orthogonal projection onto $\mathcal{S}_i \cap \mathcal{S}_j$ if and only if $\boldsymbol{P}_i$ and $\boldsymbol{P}_j$ commute. In the general case, $\boldsymbol{P}_{i \wedge j} = \lim_{n \to \infty} (\boldsymbol{P}_i \boldsymbol{P}_j)^n$. In Fig. 1, the intersection $\mathcal{S}_{\text{man on a boat}} = \mathrm{span}(\boldsymbol{x}_1, \boldsymbol{x}_2)$ and $\mathcal{S}_{\text{is fishing}} = \mathrm{span}(\boldsymbol{x}_1, \boldsymbol{x}_3)$ yields $\mathcal{S}_{\text{man fishing on a boat}} = \mathrm{span}(\boldsymbol{x}_1)$.

**Disjunction** ($i \vee j$). Corresponds to the span (linear sum) of subspaces: $\mathcal{S}_{i \vee j} = \mathcal{S}_i + \mathcal{S}_j$. Any vector in $\mathcal{S}_{i \vee j}$ is a linear combination of elements in $\mathcal{S}_i$ or in $\mathcal{S}_j$. For commuting subspaces, the projection onto $\mathcal{S}_{i \vee j}$ satisfies $\boldsymbol{P}_{i \vee j} = \boldsymbol{P}_i + \boldsymbol{P}_j - \boldsymbol{P}_{i \wedge j}$. In Fig. 1, the linear sum $\mathcal{S}_{\text{man fishing on a boat}} = \mathrm{span}(\boldsymbol{x}_1)$ and $\mathcal{S}_{\text{man fishing not on a boat}} = \mathrm{span}(\boldsymbol{x}_3)$ yields $\mathcal{S}_{\text{man fishing}} = \mathrm{span}(\boldsymbol{x}_1, \boldsymbol{x}_3)$.

**Complement** ($\neg i$). Corresponds to the subspace of all vectors orthogonal to the subspace: $\mathcal{S}_{\neg i} = \mathcal{S}_i^\perp$. The projection operator onto $\mathcal{S}_i^\perp$ is given by $\boldsymbol{P}_{\neg i} = \boldsymbol{I} - \boldsymbol{P}_i$. In Fig. 1, the complement of $\mathcal{S}_{\text{man on a boat}} = \mathrm{span}(\boldsymbol{x}_1, \boldsymbol{x}_2)$ is given by $\mathcal{S}_{\text{man on a boat}}^\perp = \mathrm{span}(\boldsymbol{x}_3)$.

## 3.2 REPRESENTING SUBSPACES AS SOFT PROJECTION OPERATORS

While the orthogonal projector from Eq. (1) offers a rich and interpretable parameterization of a subspace, its optimization poses a challenge for gradient-based methods. Since the rank of a subspace is integer-valued, the space of all subspaces (a union of Grassmannian manifolds) is stratified and non-differentiable across rank changes. This makes it hard to simultaneously learn orientation and dimensionality via gradient-based methods.

**Soft Projection Operators.** To overcome the challenges associated with learning adaptive-rank subspaces we introduce a relaxation of the projection operator in Eq. (1). For a rank-$r$ subspace $\mathcal{S}$ spanned by the columns of $\boldsymbol{X} \overset{\text{SVD}}{=} \boldsymbol{U}\boldsymbol{\Sigma}\boldsymbol{V}^\top \in \mathbb{R}^{d \times n}$, where $\boldsymbol{\Sigma} = \mathrm{diag}(\{\sigma_i\}_{i=1}^r)$ and $\boldsymbol{U} \in \mathbb{R}^{d \times r}$ is the orthonormal basis of $\mathcal{S}$, we define a soft projector via Tikhonov regularization

$$\tilde{\boldsymbol{P}} := \boldsymbol{X}(\boldsymbol{X}^\top \boldsymbol{X} + \lambda \boldsymbol{I})^{-1} \boldsymbol{X}^\top = \boldsymbol{U}\mathrm{diag}\left(\left\{\frac{\sigma_i^2}{\sigma_i^2 + \lambda}\right\}_{i=1}^r\right)\boldsymbol{U}^\top, \quad \lambda > 0. \tag{5}$$

Unlike a true projector ($\boldsymbol{P}^2 = \boldsymbol{P}$), $\tilde{\boldsymbol{P}}$ is a *soft projector*: its eigenvalues vary smoothly in $[0, 1)$ rather than being binary. This makes the operator differentiable with respect to both orientation and rank, avoiding hard rank jumps and enabling gradual changes in dimensionality. Geometrically, this relaxation replaces the stratified manifold of projectors with a smooth manifold of PSD operators. From a Bayesian point of view, it corresponds to a Gaussian prior with precision $\lambda \boldsymbol{I}$.

For small values of $\lambda$, the soft projectors in Eq. (5) provide accurate surrogates for the algebraic operations and metrics introduced in §3.1. The approximation error depends primarily on the weakest nonzero singular value $\sigma_r$ of $\boldsymbol{X}$, being upper bounded by (see Appendix A)

$$\epsilon(\sigma_r, \lambda) = \lambda/(\sigma_r^2 + \lambda). \tag{6}$$

Table 1: Soft approximations of projection operations derived from $\boldsymbol{X}_i$ and $\boldsymbol{X}_j$. Errors are in operator norm, except rank (relative absolute error). $\sigma_r, \eta_r$ are the weakest non-null singular values of $\boldsymbol{X}_i$ and $\boldsymbol{X}_j$. $\epsilon(\sigma_r, \lambda) = \lambda/(\sigma_r^2 + \lambda)$.

| | Projector | Negation | Intersection | Linear sum | Rank |
|---|---|---|---|---|---|
| Exact | $\boldsymbol{X}(\boldsymbol{X}\boldsymbol{X}^\top)^\dagger \boldsymbol{X}^\top$ | $\boldsymbol{I} - \boldsymbol{P}$ | $\boldsymbol{P}_i \boldsymbol{P}_j$ | $\boldsymbol{P}_i + \boldsymbol{P}_j - \boldsymbol{P}_i \boldsymbol{P}_j$ | $\mathrm{Tr}(\boldsymbol{P})$ |
| Soft | $\boldsymbol{X}(\boldsymbol{X}\boldsymbol{X}^\top + \lambda \boldsymbol{I})^{-1}\boldsymbol{X}^\top$ | $\boldsymbol{I} - \tilde{\boldsymbol{P}}$ | $\tilde{\boldsymbol{P}}_i \tilde{\boldsymbol{P}}_j$ | $\tilde{\boldsymbol{P}}_i + \tilde{\boldsymbol{P}}_j - \tilde{\boldsymbol{P}}_i \tilde{\boldsymbol{P}}_j$ | $\mathrm{Tr}(\tilde{\boldsymbol{P}})$ |
| Error | $\epsilon(\sigma_r, \lambda)$ | $\epsilon(\sigma_r, \lambda)$ | $\epsilon(\sigma_r, \lambda) + \epsilon(\eta_r, \lambda)$ | $2(\epsilon(\sigma_r, \lambda) + \epsilon(\eta_r, \lambda))$ | $\epsilon(\sigma_r, \lambda)$ |

As $\lambda \to 0$, $\epsilon(\sigma_r, \lambda) \to 0$ and we recover the orthogonal projection operator $\tilde{\boldsymbol{P}} \to \boldsymbol{P}$, while larger $\lambda$ enforces smoother, more regularized projectors. Table 1 summarizes how each operation is approximated using $\tilde{\boldsymbol{P}}$ and the resulting deviation from the orthogonal projector ($\lambda = 0$).

**Subspace Projection Head (SPH).** To bridge our subspace representations with transformer models, we introduce the *Subspace Projection Head (SPH)*. A transformer first encodes text inputs into a contextualized hidden state $\boldsymbol{H} \in \mathbb{R}^{h \times m}$ (where $m$ is sequence length, $h$ is hidden dimension). The SPH transforms this hidden state $\boldsymbol{H}$ into a fixed-size set of $n$ vectors $\boldsymbol{X} \in \mathbb{R}^{d \times n}$ that span a subspace $\mathcal{S}$ and then explicitly computes the corresponding soft projector $\tilde{\boldsymbol{P}}$.

We map the hidden state $\boldsymbol{H}$ into a sequence-length-invariant subspace in three stages. First, we augment the transformer with a set of $n$ learnable query vectors $\boldsymbol{Q} \in \mathbb{R}^{h \times n}$. These queries attend to $\boldsymbol{H}$ (acting as keys and values) via Multi-Head Attention (MHA), pooling $n$ embeddings $\boldsymbol{X}'$,

$$\boldsymbol{X}' = \mathrm{MHA}(\mathrm{query} = \boldsymbol{Q}, \mathrm{key} = \boldsymbol{H}, \mathrm{value} = \boldsymbol{H}) \in \mathbb{R}^{h \times n}. \tag{7}$$

This ensures the dimensions of $\boldsymbol{X}'$ are independent of the sequence length $m$. However, the rank of $\boldsymbol{X}'$ is still limited: since each head outputs a linear combination of the columns of $\boldsymbol{H}$, then $\mathrm{rank}(\boldsymbol{X}') \leq n_{\mathrm{heads}} \cdot \mathrm{rank}(\boldsymbol{H}) \leq m \cdot n_{\mathrm{heads}}$. We address this via a Multi-Layer Perceptron (MLP) which maps the $n$ $h$-dimensional vectors from the MHA output to $\mathbb{R}^d$ as $\boldsymbol{X} = \mathrm{MLP}(\boldsymbol{X}')$. This yields the subspace matrix $\boldsymbol{X} \in \mathbb{R}^{d \times n}$, which spans the subspace $\mathcal{S}$. Finally, the actual representation *i.e.*, the soft projector $\tilde{\boldsymbol{P}}$ onto $\mathcal{S}$, is computed from $\boldsymbol{X}$ using the closed-form in Eq. (5).

## 3.3 TRAINING METHODOLOGY

We learn subspaces end-to-end via gradient descent, requiring no special pretraining, or training constraints. Depending on the downstream task, we employ one of the following loss functions.

**Reconstruction.** For similarity-based tasks, we use an InfoNCE loss (van den Oord et al., 2019) with the subspace similarity computed via $\mathrm{sim}(\tilde{\boldsymbol{P}}_i, \tilde{\boldsymbol{P}}_j)$, from Eq. (3).

**Link Prediction.** In link prediction tasks, we optimize the normalized inclusion score $\mathrm{NIS}(\tilde{\boldsymbol{P}}_i \mid \tilde{\boldsymbol{P}}_j)$ from Eq. (4) directly and consider the margin loss (Vendrov et al., 2016)

$$L = \sum_{i,j \in \mathcal{P}} [\gamma_+ - \mathrm{NIS}(\tilde{\boldsymbol{P}}_i \mid \tilde{\boldsymbol{P}}_j)]_+ + \sum_{i,j \in \mathcal{N}} [\mathrm{NIS}(\tilde{\boldsymbol{P}}_i \mid \tilde{\boldsymbol{P}}_j) - \gamma_-]_+, \tag{8}$$

where $[\cdot]_+$ denotes the ReLU function. Here, $\gamma_+, \gamma_- \in (0, 1)$ are the positive and negative margins and $\mathcal{P}$ and $\mathcal{N}$ the set of positives and negatives, respectively.

**NLI Classification.** Textual Entailment presents a unique challenge, requiring not just a measure of inclusion but also an explicit model of neutrality. For a premise $p$ and hypothesis $h$, we model the relation $Y \in \{E, N, C\}$ (entailment, neutral, contradiction) as a discrete latent variable. For $Y \in \{E, C\}$, we assume the generative process for $S = \mathrm{NIS}(\tilde{\boldsymbol{P}}_h \mid \tilde{\boldsymbol{P}}_p)$

$$S \mid (Y = y) \sim \mathrm{Beta}(\alpha_y, \beta_y), \quad y \in \{E, C\}, \tag{9}$$

with $\alpha_y \leq \beta_y$ if $y = C$ and $\beta_y \leq \alpha_y$ if $y = E$. For neutrals, subspace inclusion does not provide a reliable signal. Instead, we model neutrality independently by an MLP as

$$P(Y = y \mid \tilde{\boldsymbol{P}}_p, \tilde{\boldsymbol{P}}_h) := \sigma\left(\mathrm{MLP}\left(\tilde{\boldsymbol{P}}_p, \tilde{\boldsymbol{P}}_h, \tilde{\boldsymbol{P}}_p \tilde{\boldsymbol{P}}_h, \tilde{\boldsymbol{P}}_h \tilde{\boldsymbol{P}}_p\right)\right), \quad y = N \tag{10}$$

Table 2: WORDNET **reconstruction**. mAP = Mean Average Precision, MR = Mean Rank, $\rho$ = Spearman correlation between taxonomy rank and subspace dimension or norm (for $\mathcal{P}^{10}, \mathcal{H}^{10}$).

| Method | Nouns | | | Verbs | | |
|---|---|---|---|---|---|---|
| | mAP ($\uparrow$) | MR ($\downarrow$) | $\rho$ ($\uparrow$) | mAP ($\uparrow$) | MR ($\downarrow$) | $\rho$ ($\uparrow$) |
| Euclidean ($\mathbb{R}^{128}$) | 95.1 | 1.31 | – | 98.6 | 1.04 | – |
| Poincaré ($\mathcal{P}^{10}$) | 86.5 | 4.02 | 58.5 | 91.2 | 1.35 | 55.1 |
| Lorentz ($\mathcal{H}^{10}$) | 92.8 | 2.95 | 59.5 | 93.3 | 1.23 | 56.6 |
| Subspaces ($SE^{128}$) | **98.6** | **1.04** | **68.0** | **99.9** | **1.00** | **67.0** |

where $\sigma(\cdot)$ denotes the sigmoid function. Assuming uniform priors for entailment and contradiction classes, conditional on non-neutrality, we compute posterior probabilities for $y = E$ and $y = C$, denoted $P(Y = y \mid S = s, Y \in \{E, C\})$. The final posterior probabilities for $y \in \{E, C\}$ are then derived by combining the MLP output for neutrality with the Beta posteriors for non-neutrality:

$$P(Y = y \mid \tilde{\boldsymbol{P}}_p, \tilde{\boldsymbol{P}}_h, S = s) = (1 - P(Y = N \mid \tilde{\boldsymbol{P}}_p, \tilde{\boldsymbol{P}}_h))P(Y = y \mid S = s, Y \neq N), \quad (11)$$

for $y \in \{E, C\}$. The posteriors in Eqs. (10) and (11) are optimized via a cross-entropy loss.

A key insight into how these losses shape the subspaces is revealed by the gradient dynamics. As derived in Appendix B, $\nabla_{\boldsymbol{X}_i} \text{sim}(\tilde{\boldsymbol{P}}_i, \tilde{\boldsymbol{P}}_j)$ encourages subspace $i$ to expand along the principal directions of subspace $j$ that it currently lacks. This update naturally promotes subspace inclusion, and the gradient vanishes once one subspace is contained within the other, leading to stable convergence.

**Efficiency Considerations.** While computing $\tilde{\boldsymbol{P}}$ from $\boldsymbol{X} \in \mathbb{R}^{d \times n}$ is $\mathcal{O}(n^3)$ in the number of vectors $n$, and has a memory footprint that scales with $d^2$, where $d$ is the ambient dimension, our approach is practical for two key reasons. First, the model learns a data-dependent rank for each subspace. As our experiments demonstrate, this allows for considerable compression of $\tilde{\boldsymbol{P}}$ via low-rank approximations. Second, the subspace similarity (3) and NIS (4) are equivalent to dot products between the vectorized matrices: $\text{Tr}(\tilde{\boldsymbol{P}}_i \tilde{\boldsymbol{P}}_j) = \text{vec}(\tilde{\boldsymbol{P}}_i)^\top \text{vec}(\tilde{\boldsymbol{P}}_j)$. This allows our subspaces to be indexed by highly optimized vector search libraries, making large-scale retrieval feasible.

## 4 EXPERIMENTS

We empirically validate our embeddings' ability to model large-scale hierarchies and textual entailment on a suite of benchmarks including WORDNET (Miller, 1995) reconstruction in §4.1 and link prediction in §4.2, HyperLex (Vulić et al., 2017), and SNLI (Bowman et al., 2015) in §4.3.

### 4.1 WORDNET RECONSTRUCTION

In WORDNET's reconstruction task, all edges from the full transitive closure of the noun and verb hypernymy hierarchies are used for training and testing. The goal is to assess the capacity of the representations to capture known hierarchical relations by providing only pairwise relations.

**Experimental Details.** Each node in the graph is represented by a soft projection matrix (5), with $\lambda = 0.2$, parameterized by a matrix $\boldsymbol{X}_i \in \mathbb{R}^{128 \times 128}$. For each training edge $(u, v)$, we sample 19 nodes $v' \neq u$ such that neither $(u, v')$ nor $(v', u)$ are in the train split and optimize InfoNCE using Adam (Kingma & Ba, 2017). During evaluation, we first compute the subspace similarity $\text{Tr}(\tilde{\boldsymbol{P}}_u \tilde{\boldsymbol{P}}_v)$ of every edge $(u, v)$ in the transitive closure. We then rank each of these scores among those of all node pairs that are not connected in the transitive closure. Based on these rankings, we report the mean rank (MR) and the mean average precision (mAP). Additional details in Appendix D.1.

**Reconstruction Results.** Our method achieves state-of-the-art performance on the WordNet reconstruction. As shown in Table 2, our subspace representations ($SE^{128}$) significantly outperform both Hyperbolic (Poincaré $\mathcal{P}^{10}$ and Lorentz $\mathcal{H}^{10}$ models), and Euclidean embeddings ($\mathbb{R}^{128}$) baselines, with a near-perfect reconstruction on the shallower verb hierarchy.

Table 3: HYPERLEX **lexical entailment** Spearman's rank correlation (WORDNET embeddings).

| | $\mathbb{R}^5$ | $\mathcal{P}^5$ | DOE-A$^{50}$ | SE$^{128}$ ($\lambda$=0.2) | SE$^{128}$ ($\lambda$=0.6) |
|---|---|---|---|---|---|
| $\rho$ ($\uparrow$) | 0.389 | 0.512 | 0.590 | 0.683 | **0.734** |

Table 4: WORDNET noun **link prediction** F1-Scores ($\uparrow$). Superscript denotes dimension.

| Non-Basic Edges | $\mathbb{R}^{10}$ | OE$^{10}$ | $\mathcal{P}^{10}$ | Cones$^{10}$ | Disk$^{10}$ | UHS$^{10}$ | SE$^{64}$ | SE$^{128}$ |
|---|---|---|---|---|---|---|---|---|
| 0% | 29.4 | 43.0 | 29.0 | 32.4 | 36.5 | 52.2 | $49.0 \pm 0.11$ | $\mathbf{53.4 \pm 0.41}$ |
| 10% | 75.4 | 69.7 | 71.5 | 84.9 | 79.5 | 89.4 | $93.6 \pm 0.06$ | $\mathbf{94.3 \pm 0.05}$ |
| 25% | 78.4 | 79.4 | 82.1 | 90.8 | 90.5 | 95.7 | $95.9 \pm 0.08$ | $\mathbf{95.9 \pm 0.11}$ |
| 50% | 78.1 | 84.1 | 85.4 | 93.8 | 94.2 | **97.0** | $95.8 \pm 0.07$ | $95.5 \pm 0.06$ |

To assess how these representations generalize to graded lexical entailment, we evaluated them on the HYPERLEX noun subset without fine-tuning (see Appendix D.3). We quantify entailment using the NIS from Eq. (4), selecting the synset pair with maximal similarity for disambiguation (Athiwaratkun & Wilson, 2018). As reported in Table 3, our embeddings demonstrate a significantly stronger correlation with human judgments than prior work. Our approach achieves a Spearman's $\rho$ of 0.73 ($\lambda = 0.6$), substantially outperforming Poincaré and Gaussian embedding baselines.

## 4.2 WORDNET LINK PREDICTION

In the link prediction task, we evaluate generalization from sparse supervision. We split the set of edges from the transitive closure that are not part the original graph (non-basic edges) into train (90%), validation (5%) and test (5%) using the data split from Suzuki et al. (2019).

**Experimental Details.** To assess how the percentage of the transitive closure seen during training impacts performance, we created partial training edge coverages by randomly sampling 0%, 10%, 25% or 50% of non-basic edges, to which we append all the basic edges. We considered two ambient space dimensions $d = 64$ and $d = 128$, setting the number of vectors as $n = d$ in each case. Training was performed by optimizing the margin loss defined in Eq. (8). During evaluation, for each positive test edge, we consider 10 negative test edges: half with a corrupted head, and half with a corrupted tail. We classify edges by thresholding the NIS from Eq. (4) and report the classification F1-Score.

**Link Prediction Results.** Link prediction results are shown in Table 4. We compare against Euclidean embeddings ($\mathbb{R}^{10}$), Order Embeddings (OE$^{10}$), Poincaré ($\mathcal{P}^{10}$) Nickel & Kiela (2017), Hyperbolic Entailment Cones (Cones$^{10}$) (Ganea et al., 2018b), Hyperbolic Disk Embeddings (Disk$^{10}$) (Suzuki et al., 2019) and the Umbral Half-Space embeddings (UHS$^{10}$) (Yu et al., 2024). Subspace embeddings SE$^{64}$ and SE$^{128}$ outperform the baselines across most supervision levels. SE$^{128}$, in particular, offers a considerable improvement when training with sparser supervision. This underscores the ability of subspace representations to infer hierarchical relations even from weak supervision.

## 4.3 SNLI

We conducted experiments on NLI using the SNLI dataset. SNLI comprises 550,152 training, and 10,000 validation/test premise ($p$) - hypothesis ($h$) pairs, each annotated with one of three labels: entailment, neutral, or contradiction. We consider two regimes: 3-way, and 2-way classification (entailment vs non-entailment). For a fair comparison, we benchmarked bi-encoder baselines, using the all-miniLM-L6-v2 and mpnet-base-v2 models with a shallow MLP classifier. We considered two variants: $\text{MLP}(p, h)$, using concatenated premise $p$ and hypothesis $h$ embeddings, and $\text{MLP}(p, h, p-h)$. In our models, we map $p$ and $h$ to soft projectors via our SPH module ($\lambda = 0.05$). All models were trained with a cross-entropy loss. Additional details are provided in Appendix E.

**Results.** As shown in Table 5, our approach consistently outperforms bi-encoder baselines. For reference, we also include two GRU-based hierarchical approaches: Order Embeddings (OE) and Hyperbolic Neural Networks (HNN) Ganea et al. (2018a), which do not model neutrality. Crucially, in the 2-way setting, our method, which relies solely on subspace inclusion, consistently outperforms the MLP baselines, with a more interpretable mechanism.

Table 5: SNLI **test accuracy**: 2-way (entailment vs non-entailment) and 3-way (+Neutral).

| Method | 2-way | 3-way |
|---|---|---|
| OE (GRU) | 88.60 | – |
| HNN (GRU) | 81.19 | – |
| all-miniLM-L6-v2 (22.7m params) | | |
| MLP($p, h$) | $90.16 \pm 0.19$ | $83.57 \pm 0.11$ |
| MLP($p, h, \Delta$) | $91.02 \pm 0.10$ | $84.89 \pm 0.21$ |
| SPH (SE$^{64}$) | $91.02 \pm 0.11$ | $84.43 \pm 0.07$ |
| SPH (SE$^{128}$) | $\mathbf{91.25 \pm 0.09}$ | $\mathbf{85.41 \pm 0.09}$ |
| mpnet-base-v2 (109m params) | | |
| MLP($p, h$) | $90.67 \pm 0.33$ | $84.10 \pm 0.27$ |
| MLP($p, h, \Delta$) | $91.74 \pm 0.07$ | $85.68 \pm 0.15$ |
| SPH (SE$^{64}$) | $91.77 \pm 0.42$ | $85.63 \pm 0.14$ |
| SPH (SE$^{128}$) | $\mathbf{91.91 \pm 0.08}$ | $\mathbf{85.80 \pm 0.05}$ |

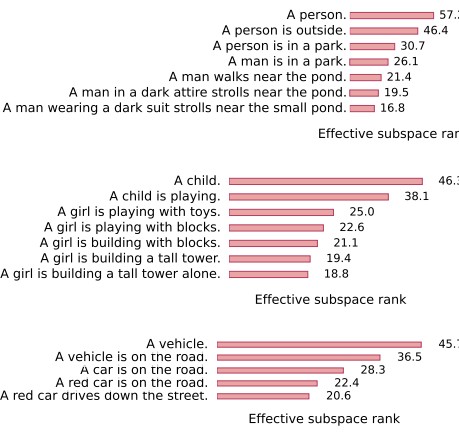

Figure 2: Example *effective subspace ranks*.

## 4.4 Composite Entailment

To quantify the logical structure of our representations, namely the ability to model conjunctions and negations, we constructed an evaluation set consisting of 600 premise-composite hypothesis pairs. Given a premise (*e.g.*, "Two children are sitting on a red picnic blanket, eating sandwiches"), we generate two conjunction-based hypotheses (*e.g.*, "People are eating" $\wedge$ "People sitting on a blanket") and two hypotheses combined via negation (*e.g.*, "People are eating" $\wedge \neg$ "People sitting directly on the grass"). The models trained on two-way SNLI from Table 5 were evaluated zero-shot. Because vectors lack native logical operators, we used vector averaging for conjunction and vector difference for negation. In contrast, subspaces were composed via intersections and complements, as in §3.1. We report AUC for conjunction and negation separately, using the NIS for our subspace embeddings and entailment probabilities for the vector baselines. Additional details in Appendix E.2

**Results** As shown in Table 7, vector baselines perform adequately on conjunction (83-91% AUC), confirming that vector averaging is a reasonable heuristic for additive semantics. However, they suffer a catastrophic failure on negations, dropping to near-random performance (49-69% AUC). In contrast, all our subspace embeddings exceed 90% AUC on both operations, retaining thus the same predictive power observed for atomic hypotheses.

## 5 Efficiency Analysis

By vectorizing the similarity (3) and the NIS (4), we can make our embeddings compatible with fast search libraries like FAISS. This contrasts with non-Euclidean embeddings requiring brute-force search. We benchmarked retrieval latency on CPU over the 155,070 Flickr30k captions (batch-size 128). The results in Table 6 show that SE$^{128}$ is nearly **8× faster** than a 10D Poincaré ($\mathcal{P}^{10}$) baseline. The encoding overhead introduced by the SPH is also minimal, averaging at an additional 0.12ms/query on GPU (Appendix G).

Table 6: Search latency.

| Latency (ms/query) | |
|---|---|
| $\mathcal{P}^{10}$ | $3.64 \pm 0.13$ |
| SE$^{128}$ | $0.47 \pm 0.02$ |

## 6 Qualitative Analysis

A key finding of our work is that our framework learns an interpretable geometry that maps the hierarchical structure of language onto the representations. We confirm this empirically on SNLI, using our SE$^{128}$ embeddings. In Fig. 3, we plot the histogram of the NIS (4) for premise-hypothesis pairs encoded with out mpnet-base-v2 (SE$^{128}$) subspace model. We observe that, for entailment this metric is concentrated towards 1, for contradictions it skews towards 0, and for neutrals it is centered around 0.5. This confirms that the NIS reflects the underlying entailment structure via subspace inclusion: each premise subspace is contained within the hypotheses subspaces it entails.

Table 7: Zero-shot composite entailment **ROC AUC**.

| | **AUC** | |
| Model | Conjunction ($\wedge$) | Negation ($\wedge\neg$) |
|---|---|---|
| all-MiniLM-L6-v2 + MLP($\boldsymbol{p}, \boldsymbol{h}$) | 86.45 | 57.68 |
| all-MiniLM-L6-v2 + MLP($\boldsymbol{p}, \boldsymbol{h}, \boldsymbol{p} - \boldsymbol{h}$) | 91.22 | 48.62 |
| mpnet-base-v2 + MLP($\boldsymbol{p}, \boldsymbol{h}$) | 82.91 | 55.75 |
| mpnet-base-v2 + MLP($\boldsymbol{p}, \boldsymbol{h}, \boldsymbol{p} - \boldsymbol{h}$) | 90.20 | 68.69 |
| all-MiniLM-L6-v2 + SPH (SE$^{64}$) | 94.68 | 90.49 |
| all-MiniLM-L6-v2 + SPH (SE$^{128}$) | 95.02 | 92.77 |
| mpnet-base-v2 + SPH (SE$^{64}$) | 95.87 | 93.89 |
| mpnet-base-v2 + SPH (SE$^{128}$) | **96.55** | **95.76** |

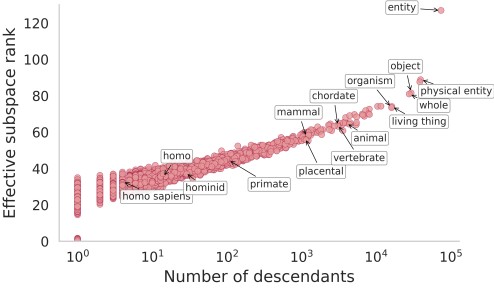

Figure 4: *Effective rank* $\mathrm{Tr}(\tilde{\boldsymbol{P}})$ vs number of descendants of WORDNET nouns.

Figure 5: Accuracy, mAP and average rank as a function of the singular value threshold.

**Rank as a Measure of Generality.** A direct consequence of this is that a subspace's *effective rank*, as measured by $\mathrm{Tr}(\tilde{\boldsymbol{P}})$, becomes an emergent measure of semantic generality. For a specific concept to be nested within many broader ones, it must occupy a lower-dimensional subspace. This property is confirmed quantitatively by the high Spearman correlation ($\rho$) between WORDNET nouns' true hierarchical positions (distance from root) and their learned *effective rank* in Table 2. We provide additional visual confirmation of this principle. Fig. 4 shows how the *effective rank* of

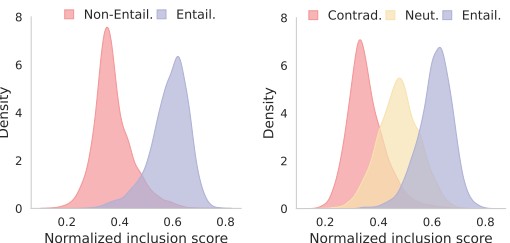

Figure 3: NIS histogram of SNLI's test set encoded with SE$^{128}$. Left: 2-way; Right: 3-way.

WORDNET's nouns grows with the number of their descendants. The annotated chain from the specific *homo sapiens* to the root noun *entity* clearly illustrates this monotonic increase. Fig. 2 shows the same phenomenon for three entailment sequences encoded with our SE$^{128}$ model. We observe again that the *effective rank* of each sentence increases as we go from a specific description to general one *e.g.*, "A red car drives down the street." $\rightarrow$ "a vehicle is on the road." $\rightarrow$ "A vehicle.".

**Dimensionality Reduction.** This learned structure, where the rank encodes specificity, makes our embeddings inherently compressible. Since each subspace dynamically allocates the dimensions needed to represent each concept, we can perform post-training compression via truncated SVD, with minimal performance loss. To assess this capability, we approximated WORDNET and SNLI embeddings $\tilde{\boldsymbol{P}}_i$ by retaining singular values greater than a threshold $\tau \in [0, 1]$ and plot the reconstruction mAP, in the case of WordNet, or the two-way accuracy, for SNLI, as well as the average subspace rank, as a function $\tau$. As shown in Fig. 5, the learned subspaces exhibit rapid spectral decay in both experiments, allowing for compression of up to $4\times$ with negligible impact on task performance. This paves the way for a new class of embeddings where representational complexity is not fixed, but a learned, data-driven property.

**Emergent Compositionality.** As evidenced by the results in Table 7, a key advantage of subspace embeddings is their emergent compositionality, which arises from the geometry of the embeddings without explicit training signals. Fig. 6 provides an example illustrating this inherent compositionality, for conjunctions $\tilde{\boldsymbol{P}}_i \tilde{\boldsymbol{P}}_j$ and negations $\boldsymbol{I} - \tilde{\boldsymbol{P}}$, in a retrieval setting. For a query formed by a logical composition of concept subspaces, we retrieve images from Flickr30k (Young et al., 2014) whose caption subspaces have the largest NIS($\tilde{\boldsymbol{P}}_{\text{query}} \mid \tilde{\boldsymbol{P}}_{\text{caption}}$). Each caption subspace is computed with our mpnet-base-v2 + SPH (SE$^{128}$) model, fine-tuned on SNLI. The results demonstrate that subspaces enable compositional retrieval, allowing for the search of novel concepts or query editing through geometric operations. Additional examples are shown in Appendix F.

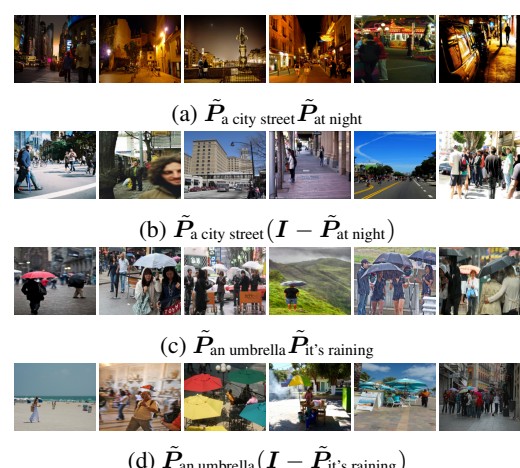

(a) $\tilde{\boldsymbol{P}}_{\text{a city street}} \tilde{\boldsymbol{P}}_{\text{at night}}$

(b) $\tilde{\boldsymbol{P}}_{\text{a city street}} (\boldsymbol{I} - \tilde{\boldsymbol{P}}_{\text{at night}})$

(c) $\tilde{\boldsymbol{P}}_{\text{an umbrella}} \tilde{\boldsymbol{P}}_{\text{it's raining}}$

(d) $\tilde{\boldsymbol{P}}_{\text{an umbrella}} (\boldsymbol{I} - \tilde{\boldsymbol{P}}_{\text{it's raining}})$

Figure 6: Flickr30k retrieval from composition of natural language queries.

## 7 LIMITATIONS

We identify two central limitations of our subspace representations and outline several promising directions for future work.

**Approximate Logic.** Our framework relies on soft projectors *i.e.*, operators with eigenvalues in $[0, 1)$, as opposed to true orthogonal projectors with binary eigenvalues. This relaxation is essential for differentiability. However, it implies that logical operations are approximate rather than symbolically exact, as evidenced by the error bounds in Table 1. Despite this, our empirical results, most prominently the NIS separation in Fig. 3 and the strong compositional performance in Table 7, suggest that these approximations remain stable and semantically coherent. A compelling direction for future work is to gradually anneal the regularization parameter $\lambda \to 0$ during training. This would allow the model to exploit soft gradients in early stages, yet increasingly sharpen its projectors toward true idempotent, orthogonal operators as training progresses, potentially recovering orthogonal projectors without sacrificing learnability.

**Storage Complexity.** A possible trade-off regarding our soft projector representations is the memory footprint, if one were to store the full $d \times d$ matrices. Encouragingly, however, the rank vs number of descendants plot in Fig. 4, as well as the dimensionality reduction plot in Fig. 5 reveal that the learned projectors exhibit an intrinsic low-rank structure, concentrating most of their semantic mass in few principal directions. This property can be exploited directly. By storing only the top-$k$ eigenvectors (and corresponding eigenvalues) of each projector, we can compress the representation significantly, bringing its memory footprint close to that of standard vector embeddings. Crucially, this compression has minimal impact on downstream tasks, suggesting that low-rank parameterizations, or even constrained low-rank training, provides a path toward scalable subspace embeddings.

## 8 CONCLUSION

This paper introduced *subspace embeddings*, a novel paradigm that addresses the limitations of vector representations in capturing logical structure and asymmetric, or hierarchical, relations. By representing concepts as subspaces, our framework naturally encodes generality through dimensionality and hierarchy through inclusion. Our evaluation across hierarchical and entailment tasks reveals the power of this inductive bias: it not only achieves state-of-the-art results but also gives rise to an emergent structure for logical composition without explicit supervision. The linearity of the core operations and metrics ensures compatibility with efficient vector search pipelines. Overall, our results establish subspace embeddings as a bridge between representation learning and logical reasoning, opening avenues for new representations that exploit the structural nature of data.

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

## A  Error Bounds for Soft Projectors

**Lemma A.1.** *Let $X = U\Sigma V^\top$, where $U \in O(d)$, and define $\tilde{P} := X(X^\top X + \lambda I)^{-1}X^\top$. We have*

$$\tilde{P} = U\Sigma^2(\Sigma^2 + \lambda I)^{-1}U^\top \tag{12}$$

*and the spectrum of $\tilde{P}$ is $\left\{\frac{\sigma_i^2}{\sigma_i^2+\lambda}\right\}_{i=1}^d$.*

*Proof.* Letting the SVD of $X$ be $U\Sigma V^\top$,

$$\begin{aligned}
\tilde{P} &= U\Sigma V^\top (V\Sigma^2 V^\top + \lambda I)^{-1}V\Sigma U^\top \\
&= U\Sigma V^\top V(\Sigma^2 + \lambda I)^{-1}V^\top V\Sigma U^\top \\
&= U\Sigma^2(\Sigma^2 + \lambda I)^{-1}U^\top.
\end{aligned} \tag{13}$$

The spectrum of $\tilde{P}$ is the diagonal of $\Sigma^2(\Sigma^2 + \lambda I)^{-1}$, which reads $\left\{\frac{\sigma_i^2}{\sigma_i^2+\lambda}\right\}_{i=1}^d$. $\square$

**Proposition A.2** (Frobenius norm error). *Let $X = U\Sigma V^\top$ be rank-$r$, where $U \in O(d)$ and let the orthogonal projector onto $\mathrm{Span}(X)$ be $P = UJ_rU^\top$. Define $\tilde{P} := X(X^\top X + \lambda I)^{-1}X^\top$. Then,*

$$\|P - \tilde{P}\|_F \leq \frac{\lambda}{\sigma_r^2 + \lambda}. \tag{14}$$

*Proof.* Let $J_r = \mathrm{BlockDiag}(I_r, \mathbf{0}_{d-r})$, where $r = \mathrm{rank}(X)$ and write the SVD of the orthogonal projector as $P = UJ_rU^\top$ for $U \in O(d)$. Using Lemma A.1, we can write $P - \tilde{P} = U(J_r - \Sigma^2(\Sigma^2 + \lambda I)^{-1})U^\top$. The Frobenius norm is invariant to orthogonal transformations $U$, hence

$$\|P - \tilde{P}\|_F^2 = \|J_r - \Sigma^2(\Sigma^2 + \lambda I)^{-1}\|_F^2 = \sum_{i=1}^r \left(1 - \frac{\sigma_i^2}{\sigma_i^2 + \lambda}\right)^2 \leq r\left(\frac{\lambda}{\sigma_r^2 + \lambda}\right)^2. \tag{15}$$

Therefore, $\|P - \tilde{P}\|_F \leq \frac{\lambda\sqrt{r}}{\sigma_r^2+\lambda}$. $\square$

**Proposition A.3** (Operator norm error). *Let $X = U\Sigma V^\top$ be rank-$r$, where $U \in O(d)$, and let the orthogonal projector onto $\mathrm{Span}(X)$ be $P = UJ_rU^\top$. Define $\tilde{P} := X(X^\top X + \lambda I)^{-1}X^\top$. Then,*

$$\|P - \tilde{P}\|_2 = \frac{\lambda}{\sigma_r^2 + \lambda}. \tag{16}$$

*Proof.* Let $J_r = \mathrm{BlockDiag}(I_r, \mathbf{0}_{d-r})$, where $r = \mathrm{rank}(X)$ and write the SVD of the orthogonal projector as $P = UJ_rU^\top$ for $U \in O(d)$. Using Lemma A.1, we can write $P - \tilde{P} = U(J_r - \Sigma^2(\Sigma^2 + \lambda I)^{-1})U^\top$. The operator norm is invariant to orthogonal transformations $U$, hence

$$\|P - \tilde{P}\|_2 = \|J_r - \Sigma^2(\Sigma^2 + \lambda I)^{-1}\|_2^2 = \max\left\{1 - \frac{\sigma_i^2}{\sigma_i^2 + \lambda}\right\}_{i=1}^r = \frac{\lambda}{\sigma_r^2 + \lambda}. \tag{17}$$

Therefore, $\|P - \tilde{P}\|_2 = \frac{\lambda}{\sigma_r^2+\lambda}$. $\square$

**Corollary A.4** (Negation operator error). *Let $X$, $P$ and $\tilde{P}$ be in the conditions of Proposition A.2. Then,*

$$\|(I - P) - (I - \tilde{P})\|_2 \leq \frac{\lambda}{\sigma_r^2 + \lambda}. \tag{18}$$

*Proof.* Note that $\|(I - P) - (I - \tilde{P})\|_2 = \|P - \tilde{P}\|_2$ and apply Proposition A.3. $\square$

**Proposition A.5** (Trace error). *Let $X = U\Sigma V^\top$ be rank-$r$, where $U \in O(d)$, and let the orthogonal projector onto $\mathrm{Span}(X)$ be $P = UJ_rU^\top$. Define $\tilde{P} := X(X^\top X + \lambda I)^{-1}X^\top$. Then,*

$$\left|\mathrm{Tr}(P) - \mathrm{Tr}(\tilde{P})\right| \leq \frac{\lambda r}{\sigma_r^2 + \lambda} \tag{19}$$

*Proof.* First write $\boldsymbol{P} - \tilde{\boldsymbol{P}} = \boldsymbol{U}(\boldsymbol{J}_r - \boldsymbol{\Sigma}^2(\boldsymbol{\Sigma}^2 + \lambda\boldsymbol{I})^{-1})\boldsymbol{U}^\top$. We have then,

$$\left|\mathrm{Tr}(\boldsymbol{P}) - \mathrm{Tr}(\tilde{\boldsymbol{P}})\right| = \left|\mathrm{Tr}(\boldsymbol{J}_r - \boldsymbol{\Sigma}^2(\boldsymbol{\Sigma}^2 + \lambda\boldsymbol{I})^{-1})\right| = \sum_{i=1}^r \left(1 - \frac{\sigma_i^2}{\sigma_i^2 + \lambda}\right) \le \frac{\lambda r}{\sigma_r^2 + \lambda}. \tag{20}$$

$\square$

**Corollary A.6** (Subspace rank error). *Letting $r := \mathrm{rank}(\boldsymbol{X})$, the relative error of estimating $r$ via $\mathrm{Tr}(\tilde{\boldsymbol{P}})$ verifies*

$$\frac{\left|\mathrm{Tr}(\tilde{\boldsymbol{P}}) - r\right|}{r} \le \frac{\lambda}{\sigma_r^2 + \lambda}. \tag{21}$$

*Proof.* Suffices to note that $r = \mathrm{Tr}(\boldsymbol{P})$ and use Proposition A.5. $\square$

**Proposition A.7** (Subspace similarity error). *Let $\boldsymbol{X}_i$ and $\boldsymbol{X}_j$ be rank-$r_i$ and $r_j$ matrices, with singular values $\{\sigma_k\}_{k=1}^d$ and $\{\eta_k\}_{k=1}^d$ (in descending order), respectively. Denote by $\boldsymbol{P}_i, \tilde{\boldsymbol{P}}_i$ and $\boldsymbol{P}_j, \tilde{\boldsymbol{P}}_j$ the respective orthogonal and soft projectors. Then,*

$$\left|\mathrm{Tr}(\boldsymbol{P}_i\boldsymbol{P}_j) - \mathrm{Tr}(\tilde{\boldsymbol{P}}_i\tilde{\boldsymbol{P}}_j)\right| \le \sqrt{r_i r_j}\left(\frac{\lambda}{\sigma_r^2 + \lambda} + \frac{\lambda}{\eta_r^2 + \lambda}\frac{\sigma_r^2}{\sigma_r^2 + \lambda}\right) \tag{22}$$

*Proof.* We have

$$\left|\mathrm{Tr}(\boldsymbol{P}_i\boldsymbol{P}_j) - \mathrm{Tr}(\tilde{\boldsymbol{P}}_i\tilde{\boldsymbol{P}}_j)\right| = \left|\mathrm{Tr}((\boldsymbol{P}_i - \tilde{\boldsymbol{P}}_i)\boldsymbol{P}_j) + \mathrm{Tr}((\boldsymbol{P}_j - \tilde{\boldsymbol{P}}_j)\tilde{\boldsymbol{P}}_i)\right|$$

$$\le \left|\mathrm{Tr}((\boldsymbol{P}_i - \tilde{\boldsymbol{P}}_i)\boldsymbol{P}_j)\right| + \left|\mathrm{Tr}((\boldsymbol{P}_j - \tilde{\boldsymbol{P}}_j)\tilde{\boldsymbol{P}}_i)\right|. \tag{23}$$

Apply Cauchy-Schwartz to both terms, we arrive at

$$\left|\mathrm{Tr}(\boldsymbol{P}_i\boldsymbol{P}_j) - \mathrm{Tr}(\tilde{\boldsymbol{P}}_i\tilde{\boldsymbol{P}}_j)\right| \le \|\boldsymbol{P}_i - \tilde{\boldsymbol{P}}_i\|_F\|\boldsymbol{P}_j\|_F + \|\boldsymbol{P}_j - \tilde{\boldsymbol{P}}_j\|_F\|\tilde{\boldsymbol{P}}_i\|_F \tag{24}$$

and we can replace $\sqrt{r_j} = \|\boldsymbol{P}_j\|_F$, $\sqrt{r_i} = \|\boldsymbol{P}_i\|_F$ and employ Proposition A.2,

$$\left|\mathrm{Tr}(\boldsymbol{P}_i\boldsymbol{P}_j) - \mathrm{Tr}(\tilde{\boldsymbol{P}}_i\tilde{\boldsymbol{P}}_j)\right| \le \|\boldsymbol{P}_i - \tilde{\boldsymbol{P}}_i\|_F\sqrt{r_j} + \|\boldsymbol{P}_j - \tilde{\boldsymbol{P}}_j\|_F\|\tilde{\boldsymbol{P}}_i\|_F$$

$$\le \sqrt{r_i r_j}\frac{\lambda}{\sigma_r^2 + \lambda} + \sqrt{r_j}\frac{\lambda}{\eta_r^2 + \lambda}\|\tilde{\boldsymbol{P}}_i\|_F. \tag{25}$$

Finally, note that $\|\tilde{\boldsymbol{P}}_i\|_F \le \sqrt{r_i}\left(\frac{\sigma_r^2}{\sigma_r^2 + \lambda}\right)$

$$\left|\mathrm{Tr}(\boldsymbol{P}_i\boldsymbol{P}_j) - \mathrm{Tr}(\tilde{\boldsymbol{P}}_i\tilde{\boldsymbol{P}}_j)\right| \le \sqrt{r_i r_j}\left(\frac{\lambda}{\sigma_r^2 + \lambda} + \frac{\lambda}{\eta_r^2 + \lambda}\frac{\sigma_r^2}{\sigma_r^2 + \lambda}\right). \tag{26}$$

$\square$

**Proposition A.8** (Intersection operator error). *Let $\boldsymbol{X}_i$ and $\boldsymbol{X}_j$ be rank-$r_i$ and $r_j$ matrices, with singular values $\{\sigma_k\}_{k=1}^d$ and $\{\eta_k\}_{k=1}^d$ (in descending order), respectively. Denote by $\boldsymbol{P}_i, \tilde{\boldsymbol{P}}_i$ and $\boldsymbol{P}_j, \tilde{\boldsymbol{P}}_j$ the respective orthogonal and soft projectors. Then,*

$$\|\boldsymbol{P}_i\boldsymbol{P}_j - \tilde{\boldsymbol{P}}_i\tilde{\boldsymbol{P}}_j\|_2 \le \frac{\lambda}{\sigma_r^2 + \lambda} + \frac{\lambda}{\eta_r^2 + \lambda}. \tag{27}$$

*Proof.* From writing $\boldsymbol{P}_i\boldsymbol{P}_j - \tilde{\boldsymbol{P}}_i\tilde{\boldsymbol{P}}_j = (\boldsymbol{P}_i - \tilde{\boldsymbol{P}}_i)\boldsymbol{P}_j + (\boldsymbol{P}_j - \tilde{\boldsymbol{P}}_j)\tilde{\boldsymbol{P}}_i$ and applying the triangle inequality

$$\|\boldsymbol{P}_i\boldsymbol{P}_j - \tilde{\boldsymbol{P}}_i\tilde{\boldsymbol{P}}_j\|_2 = \|(\boldsymbol{P}_i - \tilde{\boldsymbol{P}}_i)\boldsymbol{P}_j + (\boldsymbol{P}_j - \tilde{\boldsymbol{P}}_j)\tilde{\boldsymbol{P}}_i\|_2$$

$$\le \|\boldsymbol{P}_i - \tilde{\boldsymbol{P}}_i\|_2\|\boldsymbol{P}_j\|_2 + \|\boldsymbol{P}_j - \tilde{\boldsymbol{P}}_j\|_2\|\tilde{\boldsymbol{P}}_i\|_2. \tag{28}$$

Noting that $\|\boldsymbol{P}_j\|_2 \le 1$ and $\|\tilde{\boldsymbol{P}}_i\|_2 \le 1$ and using Proposition A.3, we have

$$\|\boldsymbol{P}_i\boldsymbol{P}_j - \tilde{\boldsymbol{P}}_i\tilde{\boldsymbol{P}}_j\|_2 \le \frac{\lambda}{\sigma_r^2 + \lambda} + \frac{\lambda}{\eta_r^2 + \lambda}. \tag{29}$$

$\square$

## B   GRADIENTS OF SOFT PROJECTION MATRICES

To understand how gradient-based training inherently shapes subspaces, we analyze the gradient flow of the subspace intersection. This reveals how projection operators evolve by incorporating missing dimensions from positive samples and repelling those aligned with negative ones.

The gradient of $\mathrm{Tr}(\tilde{P}_i \tilde{P}_j)$ with respect to $X_i$ can be derived from the identity

$$
\nabla_X \mathrm{Tr}\left((A + X^\top X X)^{-1}(X^\top B X)\right) =
$$
$$
-2CX(A + X^\top CX)^{-1}X^\top BX(A + X^\top CX)^{-1} + 2BX(A + X^\top CX)^{-1}. \tag{30}
$$

We have then

$$
\nabla_{X_i} \mathrm{Tr}\left(\tilde{P}_i \tilde{P}_j\right) = 2(I - \tilde{P}_i)\tilde{P}_j X_i(X_i^\top X_i + \lambda I)^{-1}
$$
$$
\propto \underbrace{\tilde{P}_i^\perp \tilde{P}_j}_{\text{New information}} \underbrace{X_i(X_i^\top X_i + \lambda I)^{-1}}_{\text{Spectral scaling}}. \tag{31}
$$

The spectral scaling factor $X_i(X_i^\top X_i + \lambda I)^{-1}$ acts as low-pass filter on $X_i$. If we write the SVD of $X_i$ as $X_i = U\Sigma V^\top$, then $X_i(X_i^\top X_i + \lambda I)^{-1} = U\Sigma(\Sigma^2 + \lambda I)^{-1}V^\top$. As a result, high-energy directions (associated with large singular values) are attenuated, while low-energy directions are amplified. This ensures that updates to $X_i$ preserve dominant, well-supported directions while adapting underrepresented ones.

The component $\tilde{P}_i^\perp \tilde{P}_j$, where $\tilde{P}_i^\perp = I - \tilde{P}_i$, indicates that gradient flow occurs only along directions present in subspace $j$ but orthogonal to subspace $i$, formally, in $\mathrm{range}(\tilde{P}_j) \cap \mathrm{null}(\tilde{P}_i)$. Thus, the learning signal drives $X_i$ to incorporate directions it lacks but that are represented by $X_j$, encouraging alignment without redundancy. If subspace $j$ is already contained within subspace $i$ i.e., $\tilde{P}_j \leq \tilde{P}_i$, the gradient vanishes since $\tilde{P}_j \tilde{P}_i = \tilde{P}_j$ implies $(I_d - \tilde{P}_i)\tilde{P}_j = 0$. This update mechanism shares similarities with Oja's rule in online PCA, promoting efficient subspace adaptation.

Conversely, negative pairs induce repulsive gradients, driving $X_i$ to remove directions aligned with $X_j$ and thus promoting subspace separation. Consequently, the effective dimensionality of subspace $i$ naturally adapts to encompass the union of all its relevant positive neighbors i.e.,

$$
\mathrm{rank}(\tilde{P}_i) \geq \dim \mathrm{span}\left(\bigcup_{j \in \mathrm{Pos}(i)} \mathrm{range}(\tilde{P}_j)\right). \tag{32}
$$

In other words, examples with more diverse positive neighborhoods require richer subspaces, while simpler ones can be encoded more compactly.

## C   LOGICAL COMPOSITION

In this section, we present in more detail how to construct logical queries with subspace operations.

**Conjunction.**   We approximate the projection operator of subspace intersection $\mathcal{S}_i \cap \mathcal{S}_j$ as $\tilde{P}_i \tilde{P}_j$. Recall that $P_i P_i$ is the projector onto the intersection if and only if $P_i$ and $P_j$ commute i.e., $P_i P_j = P_j P_i$. While this is not enforced in our training pipeline, we observed good empirical results from approximating the intersection operator by $\tilde{P}_i \tilde{P}_j$.

**Negation.**   We approximate the projection operator onto the subspace complement $\mathcal{S}_i^\perp$ as $I - \tilde{P}_i$. The approximation error in this case only comes from the regularization $\lambda$.

**Disjunction.**   We approximate the projection operator onto the subspace sum $\mathcal{S}_i + \mathcal{S}_j$ by explicitly building the linear sum from the basis of each soft projector. Letting the SVD of the projectors be $\tilde{P}_i = U_i \Sigma_i V_i^\top$ and $\tilde{P}_j = U_j \Sigma_j V_j^\top$, we approximate the soft projector for the subspace sum as $X(X^\top X + \lambda I)^{-1}X^\top$, where $X = \begin{bmatrix} U_i \Sigma_i^{\frac{1}{2}} & U_j \Sigma_j^{\frac{1}{2}} \end{bmatrix}$. We used the same $\lambda$ as the one used during training.

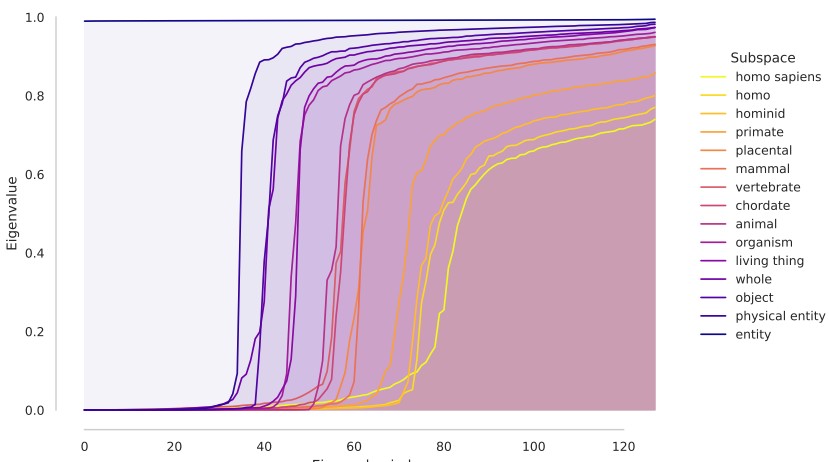

Figure 7: Sorted eigenvalues of the soft projection operators $\tilde{P}$ for nouns in the hypernymy chain *homo sapiens → entity*. As we move from specific to general concepts, the subspace's effective rank gradually expands. This illustrates how our soft projectors naturally capture concept specificity: specific nouns like *homo sapiens* activate fewer dimensions (eigenvalues near zero), while broader concepts like *entity* activate more dimensions (eigenvalues near one).

# D  WORDNET EXPERIMENTS

WORDNET's noun hierarchy has 82,115 nodes and 75,850 edges. The verb hierarchy is smaller, featuring 13,767 nodes and 13,239 edges. Their transitive closures are significantly denser, with 663,508 (noun) and 35,079 (verb) edges. All WORDNET experiments were conducted on a RTX8000 GPU with 49GB of memory.

## D.1  RECONSTRUCTION

**Experimental Details.** We parameterize each node's subspace with a matrix $X_i \in \mathbb{R}^{128 \times 128}$, initialized with entries from a zero-mean Gaussian distribution with standard deviation 0.0001. The regularizer was set $\lambda = 0.2$. For each training edge $(u, v)$, we sample 19 nodes $v' \neq u$ such that neither $(u, v')$ nor $(v', u)$ are in the train split and optimized InfoNCE, applying the the subspace similarity $\text{Tr}(\tilde{P}_i \tilde{P}_j)$ from Eq. (3) to soft projectors. We used Adam Kingma & Ba (2017), with a batch-size of 128 and learning rate of 0.0005. During evaluation, we compute the similarity $\text{Tr}(\tilde{P}_u \tilde{P}_v)$ of each edge $(u, v)$ in the full transitive closure $\text{TC}(\mathcal{G})$ and rank it among the those of all node pairs that are not connected in the transitive closure $\{\text{Tr}(\tilde{P}_u \tilde{P}_{v'}) : (u, v') \notin \text{TC}(\mathcal{G})\}$.

**Visualization of the Spectrum of WordNet Nouns.** In Fig. 7, we plot the sorted eigenvalues of our soft projector representations ($\tilde{P}$) for WORDNET nouns, traversing a hypernymy chain from *homo sapiens* to *entity*. This plot illustrates two key properties:

- Smooth Eigenvalue Distribution: Unlike the binary (0 or 1) eigenvalues of orthogonal projection matrices, the eigenvalues of $\tilde{P}$ are smooth within [0,1]. This smoothness is crucial for our learnable, soft subspace representations.

- Effective Rank Justification: The plot directly justifies our use of $\text{Tr}(\tilde{P})$ as a measure of the *effective rank* of a concept's subspace. For orthogonal projection operators, the trace (sum of eigenvalues) precisely equals the subspace's rank due to their binary eigenvalues. Here, while eigenvalues are not binary, the plot clearly shows the distribution of activated dimensions for each concept. For instance, the broad concept *entity* utilizes all 128 dimensions, with all eigenvalues near one. In contrast, *homo sapiens* activates fewer dimensions, with most eigenvalues approaching zero.

Table 8: **SNLI test accuracy** for all-miniLM-L6-v2 + SPH ($SE^{128}$) across different values of the regularization hyperparameter $\lambda$.

|  | $\lambda$ | | | |
|---|---|---|---|---|
|  | 0.01 | 0.05 | 0.1 | 0.2 |
| **2-way** | 91.18 | 91.26 | 91.12 | 91.06 |
| **3-way** | 85.27 | 85.34 | 85.61 | 85.62 |

## D.2 LINK PREDICTION

**Experimental Details.** For link prediction, every node is initialized as a random matrix $\boldsymbol{X}_i^{d \times n}$, with entries sampled from a zero-mean Gaussian distribution ($\sigma = 0.0001$). In our experiments we considered $d = n = 64$ as well as $d = n = 128$. The soft projector regularizer was set to $\lambda = 0.2$. We optimized the margin loss from Eq. (8) with $\gamma_+ = 0.9$ and $\gamma_- = 0.5$ for 0% of non-basic edges, and $\gamma_+ = 0.8$, $\gamma_- = 0.1$ for the remaining percentages. To compute this loss, we used 10 negatives per each observed positive edge $(u, v)$. Negatives were generated by sampling 5 corrupted-tail $(u, v')$ and 5 corrupted-head $(u', v)$ examples per positive edge, with corrupted nodes sampled from the entire set of nodes. The results were averaged over 5 random seeds, employing Adam (Kingma & Ba, 2017) with a constant learning rate of 0.0005 and a batch-size of 128 to perform the optimization.

## D.3 GRADED LEXICAL ENTAILMENT

**Experimental Details.** For our HYPERLEX experiment, we use the noun subset (2,163 pairs), which provides human-annotated scores (0-10) for word pairs $(u, v)$, quantifying the degree to which $u$ is a type of $v$. We quantify entailment using the NIS from Eq. (4), with word sense disambiguation performed as in Athiwaratkun & Wilson (2018), by selecting the WORDNET synset pair with maximal subspace similarity $\text{Tr}(\tilde{\boldsymbol{P}}_i, \tilde{\boldsymbol{P}}_j)$.

# E  NLI EXPERIMENTS

**Experimental Details.** All experiments utilized a maximum sequence length of 35. We trained all-MiniLM-L6-v2 and all-mpnet-base-v2 with a batch size of 1024. Optimization was performed using Adam (Kingma & Ba, 2017), employing a learning rate of 0.0001 and no weight decay. An exponential learning-rate scheduler with a gamma of 0.9 was used. For the MLP-based baselines, premise and hypothesis embeddings were first computed by mean pooling the transformer's output hidden state before being passed to the MLP classification head. The MLP classification head consisted of 3 layers, featuring LeakyReLU activations and matching the hidden dimension of its corresponding transformer. A label smoothing of 0.1 was consistently applied across all training runs. The Beta priors of our model were initialized as ($\alpha_C = 1$, $\beta_C = 6$) and ($\alpha_E = 6$, $\beta_E = 1$) and were optimized during training. All experiments were averaged over 5 random seeds on a RTX8000 GPU with 49GB of memory.

## E.1 SENSITIVITY TO THE REGULARIZATION HYPERPARAMETER ($\lambda$)

Table 8 reports SNLI classification accuracy for the all-miniLM-L6-v2 + SPH ($SE^{128}$) model for several choices of the regularization parameter $\lambda > 0$. Recall that $\lambda$ controls the amount of spectral smoothing applied to the projection operator: when $\lambda = 0$ the operator reduces to an orthogonal projector, while $\lambda > 0$ yields a PSD operator with eigenvalues in $[0, 1)$, avoiding a binary spectrum and ensuring differentiability. Across the tested range, model accuracy varies only marginally, indicating that the method remains performant.

## E.2 COMPOSITE ENTAILMENT

**Dataset.** We start with a set of 150 premises. For each premise, we define two entailed atomic hypotheses: $h_1$ and $h_2$. We define a third atomic hypothesis $h_3$ (contradicted) that shares context

with $h_2$ but is factually incompatible with $p$. In total, we have thus 300 premise-hypotheses pairs featuring hypotheses combined via conjunction, and another 300 pairs with hypotheses combined via conjunction and negation. For each $p$ we have then:

- **Entailed hypotheses**: $h_1 \wedge h_2$ and $h_1 \wedge \neg h_3$ remain entailed by $p$.

- **Contradicted hypotheses**: $h_1 \wedge h_3$ and $h_1 \wedge \neg h_2$ contradict $p$ due to the inclusion of $h_3$ in one case, and the negation of $h_2$ in the other.

We filtered for examples where atomic entailments were correctly predicted by the baselines, ensuring that failures in the composite task were due to the composition operation itself, not a failure to represent the individual atomic sentences. We have, for example,

- **Premise:** "Two children are sitting on a red picnic blanket, eating sandwiches.".

- **Entailed:** "People are eating." $\wedge$ "People sitting on a blanket.", "People are eating." $\wedge \neg$ "People sitting directly on the grass".

- **Contradicted:** "People are eating." $\wedge$ "People sitting directly on the grass.", "People are eating." $\wedge \neg$ "People sitting on a blanket.".

## F   FLICKR30K RETRIEVAL WITH COMPOSITE QUERIES

To qualitatively demonstrate the zero-shot compositional capabilities of our representations, we utilize the SNLI-fine-tuned mpnet-base-v2 + SPH (SE$^{128}$) model.

**Methodology.** We embed the 155,070 Flickr30k captions to serve as the candidate retrieval pool. We construct composite queries by encoding individual phrases and combining them via logical subspace operations. For instance, to represent "a dog running" AND "on the beach":

1. Compute the soft projector for the first phrase: $\tilde{P}_{\text{a dog running}}$.

2. Compute the soft projector for the second phrase: $\tilde{P}_{\text{on a beach}}$.

3. Formulate a composite query by the composition of $\tilde{P}_{\text{a dog running}}$ and $\tilde{P}_{\text{on a beach}}$.

4. Rank all candidate captions in the Flickr30k corpus against this composite subspace $\tilde{P}_{\text{query}}$ using the Normalized Inclusion Score (NIS) defined in Eq. (4)

**Results.**   In Fig. 8, we display the ground-truth images associated with the top-6 retrieved captions for 20 different composite queries. These results illustrate that the model successfully retrieves instances satisfying multiple logical constraints simultaneously, despite being trained solely on textual entailment pairs.

## G   EFFICIENCY EXPERIMENTS

**Training Time and Peak GPU Memory.**   Using SNLI with a batch size of 1024 on a single RTX8000 GPU, Table 9 summarizes the wall-clock time per epoch and peak GPU memory for both our model and for the vector baselines used in the NLI experiments. Our training incurs a moderate overhead: for the larger mpnet-base-v2 backbone, training our SE$^{128}$ model is approximately $1.3\times$ slower and uses $1.2\times$ more memory than the vector baseline.

**Encoding Time.**   We measured the overhead introduced by our SPH module when encoding Flickr30k captions on a RTX8000 GPU with 49GB of memory. To isolate the computation cost of the forward pass, tokenization (max-size of 35) and data transfers were computed beforehand. Table 10 shows the average encoding time per query (forward-pass) for different batch sizes, demonstrating that the additional computational cost is modest, averaging at an additional 0.12ms/query for a batch size of 128.

Table 9: **Training wall-clock time (s/epoch) and peak GPU memory (MB)**. Averaged over 3 runs.

| Model | Wall-clock time (s/epoch) | Peak GPU mem. (MB) |
|---|---|---|
| all-MiniLM-L6-v2 + MLP($p, h, p - h$) | $124.63 \pm 0.88$ | $7617.95 \pm 0.04$ |
| all-MiniLM-L6-v2 + SPH (SE$^{128}$) | $218.19 \pm 1.54$ | $12633.51 \pm 0.04$ |
| mpnet-base-v2 + MLP($p, h, p - h$) | $551.07 \pm 1.45$ | $32327.88 \pm 0.15$ |
| mpnet-base-v2 + SPH (SE$^{128}$) | $710.99 \pm 3.12$ | $39399.39 \pm 0.04$ |

Table 10: **GPU average encoding time (ms/query)**, averaged over Flickr30k's captions. The overhead of the SPH module is consistently small.

| Model | Batch size | | | | |
|---|---|---|---|---|---|
| | 1 | 4 | 16 | 64 | 128 |
| mpnet-base-v2 | 5.95 | 1.74 | 0.69 | 0.59 | 0.56 |
| mpnet-base-v2 + SPH (SE$^{128}$) | 6.80 | 2.13 | 0.86 | 0.73 | 0.68 |

**Retrieval Latency.** Given the embeddings for the 155,070 captions from the Flickr30k dataset, we benchmarked top-10 retrieval latency on CPU using batches of 128 queries. We compared our subspace embeddings (SE$^{128}$) against a 10-dimensional Poincaré hyperbolic baseline ($\mathcal{P}^{10}$). Because hyperbolic distance is non-Euclidean, we applied brute-force search over the entire database, ranking by the negative hyperbolic distance. In contrast, our NIS score can be formulated as a maximum inner product search problem between query and caption vectors:

$$\text{NIS}(\tilde{\boldsymbol{P}}_{\text{caption}} \mid \tilde{\boldsymbol{P}}_{\text{query}}) = \left( \frac{\text{vec}(\tilde{\boldsymbol{P}}_{\text{caption}})}{\text{Tr}(\tilde{\boldsymbol{P}}_{\text{caption}})} \right)^{\top} \text{vec}(\tilde{\boldsymbol{P}}_{\text{query}}). \tag{33}$$

This formulation allows us therefore to use fast approximate search libraries. We indexed the normalized caption vectors using a CPU index from the FAISS library Douze et al. (2025). We used an inverted file index with Product Quantization (IndexIVFPQ). The index was trained on 50,000 vectors. We used 64 subquantizers for PQ with 8 bits per subquantizer, and set the search-time parameter to $n_{\text{probe}} = 32$.

## H    LARGE LANGUAGE MODELS

The authors are solely responsible for the research ideas, experimental design, and analysis presented in this work. Large language model (LLMs) was used for editorial assistance to enhance the paper's clarity and readability, with its contributions limited to grammar, sentence structure, and flow.

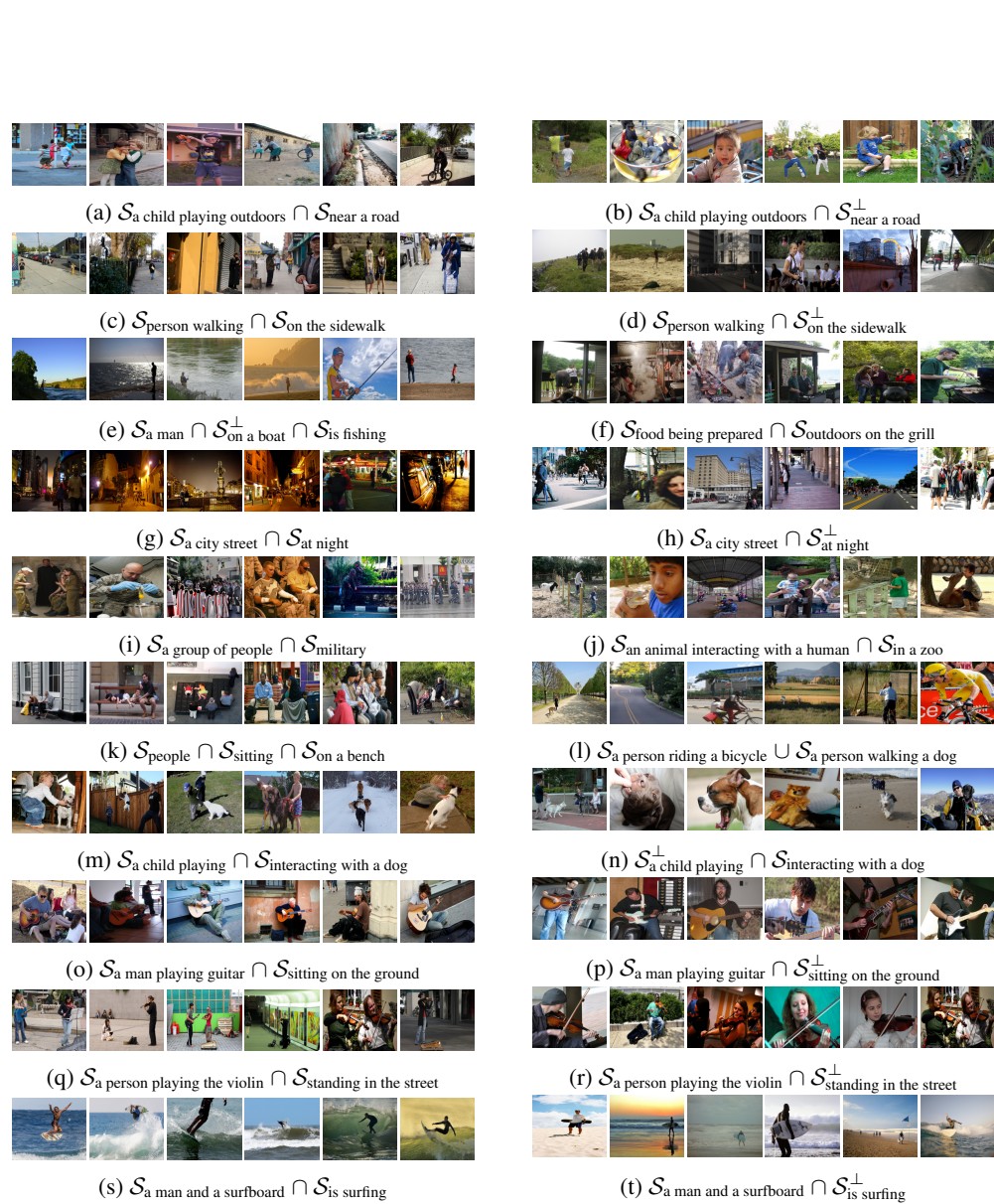

Figure 8: Each subfigure demonstrates the inherent capacity of subspace embeddings for logical composition. Queries are formed by applying subspace operations intersection ($\cap$), linear sum ($+$) and orthogonal complement ($\perp$) to the subspace embeddings of phrases or sentences, embedded by our SNLI-fine-tuned mpnet-base-v2 + SPH ($SE^{128}$) model. For each composite query, we retrieve the top Flickr30k images whose captions have the highest NIS with the composite query subspace.

