# OpenReview forum: "Native Logical and Hierarchical Representations with Subspace Embeddings"
_ICLR.cc/2026/Conference — Submitted to ICLR 2026_

### Official Review · Reviewer_kMKP · 2025-10-25

**Soundness:** 3
**Presentation:** 3
**Contribution:** 3
**Rating:** 4
**Confidence:** 4

**Summary:**

The authors proposed a fundamental shift in embedding methodology: instead of using standard vector representations, one should represent concepts as a linear subspace. They claim this constitutes a paradigm shift in embedding. Specifically, their approach constructs a subspace embedding using orthogonal projections. The authors then attempted to demonstrate that these resulting subspace embeddings exhibit emergent properties regarding logical operations, backing this significant claim with empirical geometrical analysis and experiments.

The motivation behind this paper stems from the fact that dense vector embeddings in Euclidean spaces lack in terms of capturing the directionality and hierarchical relationship among the embedded objects. Furthermore, vector spaces lack any native operators for logical conjunction and negation.

This paper proposes an alternative where instead of mapping a concept to a single vector, it is embedded as a linear subspace of $\mathbb{R}^d$. In this framework, generality and specificity of concepts get captured through subspace dimensions. The paper defines all the necessary mathematical framework for this proposed new approach. Lastly, they demonstrate the efficacy of the proposed approach for three different tasks - WordNet reconstruction, WordNet Link prediction, and SNLI.

While motivation is fully justified and the proposed approach is also novel but my fear is that almost identical conceptual framework was proposed in a series of papers related to Quantum Embeddings couple of years back. I would like authors to have a careful look at these because the ideas proposed in these papers is quite closely matching what is proposed here.

- https://papers.nips.cc/paper_files/paper/2019/file/cb12d7f933e7d102c52231bf62b8a678-Paper.pdf
- https://proceedings.neurips.cc/paper_files/paper/2020/file/b87039703fe79778e9f140b78621d7fb-Paper.pdf

Authors needs to highlight a very clear and strong differentiation as well as justification about their work in light of the above prior art.

**Strengths:**

- The motivation behind the paper and proposed idea is novel but unfortunately, almost identical idea was proposed as part of Quantum Embedding work couple of years back.
- The idea of soft projection operator is nice and gives an handle to manage non-differentiability of projection operator.
- The metric NIS defined to quantify the subspaces similarity is also nice.

**Weaknesses:**

1. I find it hard to agree with the claim that the proposed method—which uses orthogonal projections to derive subspace representations of language—is novel or a paradigm shift. The authors seem to have neglected a proper literature survey on hierarchical representation across language and knowledge bases. It's a well-established technique in the field to use orthogonal projections and their properties for building hierarchical language representations. See the works of 1) Garg et. al, NeurIPS 2019 on quantum embedding of knowledge for reasoning (https://papers.nips.cc/paper_files/paper/2019/file/cb12d7f933e7d102c52231bf62b8a678-Paper.pdf) , 2) Srivastava et.al NeurIPS 2020 on inductive quantum embedding (https://proceedings.neurips.cc/paper_files/paper/2020/file/b87039703fe79778e9f140b78621d7fb-Paper.pdf).

2. Authors discussed the properties about projection and its smooth approximation. However, I failed to understand clearly how they are constructing the subspace representations. May be the subsection “Subspace Projection Head” is not written clearly. What is the input, what are all the intermediate steps, and what is the output?

3. My initial interpretation is that the authors take a sequence representation from a transformer model, enclose the resulting vectors within an unbounding box, and define this as the input sequence's subspace. If this is the case, I do not understand how the system is learning an orthogonal (smooth) projection matrix for an input statement from the ground up. This learning process is essential to the claimed properties but is missing from my understanding of their proposed construction.

4. I believe the experiments are not exhaustive enough to convincingly demonstrate that their method surpasses previous state-of-the-art results. To strengthen their claims, they should take the following steps: First, to prove that the LLM's representation is not hierarchical, they must compare their results with the latest LLM. Second, they should perform more complex reasoning tasks to persuasively demonstrate the emergent properties of logical operations. Third, they should test their methods on reasoning tasks that utilize preferably a large size knowledge bases such as Wikidata (18 billion triples) or OpenStreetMap.

**Questions:**

- The section on Subspace Projection Head (SPH) is crucial section but the details are relatively less clear. This section can be elaborated a bit more in my view for conveying the idea bit more clearly. My understanding is that every piece of text would be first converted to matrix H which is then converted into matrix X whose columns span the corresponding subspace?

---

> ### Author Response · Authors · 2025-11-23
>
> We thank the reviewer for the constructive feedback. We particularly appreciate the references to Quantum Embeddings (QE/IQE) and the push for more complex reasoning tasks. We address the concerns below.
>
> ### Novelty & Differentiation from Quantum Embeddings (QE/IQE)
> We are grateful for the references to [QE] - Garg et al. (2019) and [IQE] - Srivastava et al. (2020), which we have included in the revised paper. While we share the high-level motivation of using subspaces, our mathematical framework and its resulting properties are fundamentally different:
> - **Representation.** QE separates entities (vectors) from concepts (subspaces). **Our contribution:** We unify this: instances and concepts are both subspaces, the former typically low-rank and while the latter tends to span more dimensions. Crucially, the dimensionality is learned from the data (as shown by the rank-hierarchy correlation in Figs. 2 and 4).
>
> - **Axis-Alignment vs. Arbitrary Orientation.** QE and IQE rely on axis-aligned subspaces. This implies that all subspaces share the same basis vectors, and thus concepts are defined by binary masks over this fixed basis. This restricts logical relations: two concepts can only interact if they explicitly share standard basis vectors. **Our contribution:** We learn subspaces with arbitrary orientation (via the learned parameterization $\mathbf{X}$). This allows our model to capture semantic correlations (soft overlaps) that are impossible in axis-aligned frameworks. Our subspaces can intersect at any angle, allowing for a much richer modeling of concept similarity and entailment.
>
> - **Optimization Landscape.** QE/IQE enforce strict geometric constraints (unit norms for vectors, commutativity for shared basis, inclusion between vectors and subspaces, non-trivial subspace ranks to avoid collapse, etc.). All these imply numerous training regularizations and *ad-hoc* loss terms. **Our Contribution:** Our Soft Orthogonal Projector ($\tilde{\mathbf{P}}$) is the key innovation that bridges geometric logic with deep learning. It is fully differentiable and unconstrained. We do not need to enforce logical consistency in the loss, instead, we show that logical structure emerges naturally from minimizing standard losses (InfoNCE and Cross-Entropy).
>
> Overall, while QE/IQE also operate with subspaces, our method is conceptually and practically distinct. We focus on continuous, orientation-aware, data-driven subspace learning for language. We do not impose subspace dimension (vectors vs subspaces), axis-alignment, nor do we enforce logical consistency in the loss functions, or require other training regularizations. All the properties presented in Section 5 emerge from the data.
>
> ### Clarification of the Subspace Projection Head
> We apologize for the lack of clarity. The reviewer asked if we just "enclose vectors in a unbounding box". This is not the case. We learn a latent set of vectors $\mathbf{X}$ from which the subspace is constructed. The pipeline is:
> 1. **Input:** Tokenized sequence of length $m$.
> 2. **Encoder:** Transformer outputs representations $\mathbf{H}\in\mathbb{R}^{h\times m}$. These are taken from the final layer of the transformer model.
> 3. **MHA:** We use a Multi-Head Attention (MHA) layer where $\mathbf{H}$ acts as Keys/Values, and a learnable, input-independent matrix $\mathbf{Q}\in\mathbb{R}^{h\times n}$ acts as Queries. The $\mathbf{Q}$ matrix is a parameter which is optimized with the rest of the transformer backbone. The MHA yields a matrix whose dimensions are independent of the input dimensions: $\mathbf{X}' \in \mathbb{R}^{h\times n}$.
> 4. **MLP:** The $n$ $h$-dimensional vectors of $\mathbf{X}’$ are projected to $d$-dimensions via an MLP, yielding $\mathbf{X}\in\mathbb{R}^{d×n}$.
> 5. **Soft Projector Construction:** Finally, we compute the projector analytically via the **closed-form expression** $\tilde{\mathbf{P}}=\mathbf{X}(\mathbf{X}^\top \mathbf{X}+\lambda \mathbf{I})^{-1}\mathbf{X}^\top$.
> The network learns to generate the vectors $\mathbf{X}$ that span the subspace, but the actual representation used for all downstream tasks is $\tilde{\mathbf{P}}$.

---

> ### Author Response · Authors · 2025-11-23
>
> (part 2)
>
>
> ### Comparison with Latest LLMs
> The reviewer suggests comparing our representations with the latest LLMs. We would like to emphasize the distinction between architecture and representation.
> - Large LLMs typically rely on cross-attention (cross-encoders) between all tokens to determine entailment. This allows for much greater accuracy, at the expense of interpretability.
> - Our work proposes a bi-encoder framework where the geometry of the embedding itself encodes the logic.
> - Comparing our embedding space to the inference pass of a large LLM would be a comparison of task performance rather than representation quality.
> - We believe our Rank vs. WordNet Depth analysis Fig 2 as well as Fig 4 is a more rigorous proof of hierarchical structure than LLM probing, as it shows the hierarchy is intrinsic to the geometry of our space, not just a feature of the training data.
>
> ### Testing on Large Knowledge Bases (Wikidata / OpenStreetMap)
> We appreciate the ambition of testing on large-scale Knowledge Graphs (KGs). This is a fascinating line of future work, but it falls outside the scope of the current paper, which introduces the foundational algebraic representation (soft projection operators) and its emergent properties.
>
> ### New Experiment: Quantification of the Emergent Logical Compositionality
> To address the reviewer's request for complex reasoning tasks demonstrating logical operations, we have added a **new composite entailment task in Section 4.4**.
> - **Setup.** We constructed a controlled evaluation benchmark consisting of 600 labeled logical composition doublets. For a given premise $p$ (*e.g.*, Two children are sitting on a red picnic blanket, eating sandwiches), we derive two entailed atomic hypotheses: $h_1$ (*e.g.*, People are eating) and $h_2$ (*e.g.*, People sitting on a blanket). We define a third atomic hypothesis $h_3$ (contradicted) that shares context with $h_2$ but is factually incompatible with $p$ (*e.g.*, People sitting directly on the grass). We filtered for examples where atomic entailments were correctly predicted by the baseline, ensuring that failures in the composite task were due to the composition. From these, we construct four composite hypotheses:
>   - **Entailed hypotheses:** $h_1 \land h_2$ and $h_1 \land\neg h_3$ remain entailed by $p$.
>   - **Contradicted hypotheses:** $h_1 \land h_3$ and $h_1 \land\neg h_2$ contradict $p$.
>
> - **Method:** Our model computes composite operators via the subspace operations described in the paper. We compare against bi-encoder baselines using vector averaging for conjunctions and difference for negations.
>
> - **Results:** Vector baselines perform reasonably on conjunctions (83-91% AUC). However, they suffer a catastrophic failure on negation, dropping to near-random performance (49-69% AUC). **In contrast, our subspace embeddings retain SoTA performance on both types of composite hypotheses (90-97%).**
>
>
> **Table: Zero-Shot Composite Entailment AUC** (see Table 7)
>
> | **Model**                                                                               | **AUC (Conjunctions)** | **AUC (Negations)** |
> |-------------------------------------------------------------------------------------------------------------------|-------------|-------------|
> | **mpnet-base-v2 + MLP ($\mathbf{p}, \mathbf{h}, \mathbf{p}-\mathbf{h}$)**  |   90.20    | 68.69      |
> | **(Ours) mpnet-base-v2 + SPH (SE$^{128}$)**                                                              |   **96.55**    | **95.76**      |
>
> **Note:** Section numbers have been updated to reflect the revised paper.

---

### Official Review · Reviewer_svE7 · 2025-10-30

**Soundness:** 4
**Presentation:** 3
**Contribution:** 4
**Rating:** 8
**Confidence:** 3

**Summary:**

The paper presents a new alternative to the traditional approach of representing data points as embeddings in a vector space. Specifically, the main idea is to embed concepts as linear learnable subspaces. This allows a much richer representation of concepts such as hierarchy and can natively represent logical operations such as conjunction, disjunction, etc as linear operations. The key technical contribution is the learning method to enable subspace representation of concepts where instead of fixing the embedding dimensions a set of  vectors of varying importances are learned through soft projection making subspaces flexible to select vectors as needed for their representation. To do this, they define a soft projection approximation where the dimensionality changes smoothly and is thus differentiable to be learned through gradient-based methods. The approach is also extended to transformer models and end-to-end learning for reconstruction, link prediction and NLI. A comprehensive set of experiments are performed on these tasks comparing them with state of the art methods in each.

**Strengths:**

+ Novel approach for representation learning, the idea of subspace learning naturally fits into more interpretable logical operations compared to existing approach. Given the generality of the formulation, this type of learning could turn out to be highly significant in several applications.
+ Comprehensive empirical validation that shows the generality of the approach in various tasks. The proposed method outperforms state of the art models in word net reconstruction, link prediction and NLI. The results also show the ability of the approach to represent meaningful hierarchies and logic compositionally using multimodal retrieval.

Overall, this seems to be strong paper with a novel, well-defined formalism that is general purpose and empirical results that strongly validate the claims.

**Weaknesses:**

Weakness
- Not a weakness as such but the paper does not spell out what and if there are limitations of the new representation. Scalability is one possible limitation perhaps (section 6 talks about this briefly) compared to standard embeddings. But in general, it would be nice to know about the trade-offs being made to achieve learning that is more semantically rich.

**Questions:**

Can you comment on the choice of baselines in the NLI tasks, are they considered state-of-the-art. The performance improvements in NLI seemed lower than the others. I would assume due to the nature of the subspaces it should be much better in answering logical queries such as entailment. Do you have some comments on this?

---

> ### Author Response · Authors · 2025-11-23
>
> We thank the reviewer for their encouraging assessment and for recognizing the generality and novelty of our subspace formulation. We are particularly glad that the bridge between interpretable logic and empirical performance resonated with you. Below we address the two key points raised: a discussion on the **limitations of our approach**, which we have added to the paper (Section 7) and the comment regarding the **performance margins on NLI**.
>
> ### Limitations
> We appreciate the suggestion to make the trade-offs explicit. We have added "Limitations" in **Section 7** of the revised paper, highlighting two key areas:
>
> - **Approximate Logic.** Our use of soft projectors (with eigenvalues in $[0,1)$), as opposed to true orthogonal projectors (with binary eigenvalues), is essential for end-to-end differentiability. However, this implies that logical operations are approximate rather than symbolically exact, as evidenced by the error bounds in Table 1. A promising avenue for future research is to anneal the regularization parameter $\lambda \to 0$ during training. This would allow the model to leverage soft gradients for optimization while progressively converging toward exact orthogonal projectors, thereby enabling precise symbolic logic for downstream applications.
>
> - **Storage Complexity.** A possible concern regarding our soft projector representations is the memory footprint, if one were to store the full $d\times d$ matrices. However, as demonstrated in Fig. 5, the learned subspaces are highly compressible. We can explicitly enforce this low-rank structure during inference by storing only the top-k eigenvectors, thus bringing the average memory usage close to standard vector embeddings, while preserving downstream performance.
>
> ### NLI Baselines and Performance Margins
> We strictly compared against bi-encoder baselines (where premise and hypothesis are encoded independently). We excluded cross-encoders (with full cross-attention) because: 1) cross-encoders are not designed or trained to produce embeddings; 2) representation learning is valuable on its own since it enables fast search, which is out-of-reach for cross-encoders. Comparing a representation-focused bi-encoder against a cross-encoder would obscure the contribution of the representations’ geometry.
>
> The reviewer correctly notes that while we achieve SoTA, the margin is not massive. We highlight the following:
> - **Bi-Encoder Saturation.** In the bi-encoder regime, performance on SNLI is heavily saturated. Simple vector models are already very efficient at capturing lexical overlap, which solves a large portion of the dataset.
>
> - **The "Structure" Win.** The true advantage of our method is not just the accuracy bump, but the emergent structure. Standard vector models achieve high accuracy but lack internal logical consistency (they cannot naturally handle negation or composition). Our subspace model matches or exceeds that accuracy, while simultaneously exhibiting interpretable properties: rank correlates with hierarchy, entailment corresponds to geometric inclusion, and the representations themselves allow for zero-shot logical composition. In short, for the same predictive power as vectors, we get a richer geometric structure.
>
> **Note:** Section number was updated to reflect revised paper.

---

### Official Review · Reviewer_Mkgu · 2025-10-31

**Soundness:** 2
**Presentation:** 2
**Contribution:** 1
**Rating:** 2
**Confidence:** 4

**Summary:**

This paper proposes to embed concepts as linear subspace, and introduces algebraic computation to embed set-theoretic relations between concepts, namely, intersection, union, and negation. Authors introduce a smooth relaxation of orthogonal projections to learn both subspace orientation and dimension through gradient descent, and experimented with well-known datasets, e.g., WordNet, NLI benchmarks, achieving state-of-art performances.

**Strengths:**

The proposed embedding method increases the interpretability of logical operations and concept structures in neural networks. The traditional semantics of logic and sets is explicitly represented in geometric entities.

**Weaknesses:**

This version of the proposed work still suffers from both technical and theoretical clarities. It seems that authors mix entities and propositions, and this will introduce problems. For example,  authors embed “man on a boat” as a concept and embedded as a single direction, then, map it to x1 and x2, where x1 might represent a“man on a boat that is fishing” while x2 might represent “man on a boat that is not fishing”. In this way, the concept “man on a boat” is represented by the subspace span(x1, x2). This raises the uniqueness problem. The “man” on a boat can also sit, stand, sleep, jump, ….

Authors target precisely embedding set-theoretic and logical concepts; they shall evaluate whether they can introduce determinacy and rigour into neural networks, but in experiments, they slip back into popular statistical evaluation and only achieved state-of-the-art performance.

**Questions:**

What can be called “concept”?

If the concept “man on a boat”, why not represent it as a topological relation ("on") between two the “man” subspace and a “boat” subspace?

How can we use the proposed embedding method to represent distance and orientation relations between objects?

---

> ### Author Response · Authors · 2025-11-23
>
> We thank the reviewer for their insightful comments regarding the intersection of geometric representations and set-theoretic relations. We value the formal logic perspective. However, we believe there are **critical misunderstandings** regarding the nature of our subspace representations and the unification of entities and propositions in our framework. We hope to clarify these below.
>
> ### Clarifying the Notion of “Concept”
> We thank the reviewer for raising this question, as it is central to our work. To answer "What can be called a concept?", we define our terminology as follows:
> - **Concept (Subspace).** A logical predicate or region of space *e.g.*, "a man fishing".
> - **Instance (Vector).** A specific state or realization *e.g.*, an image of a specific man fishing. This is a special case, corresponding to a rank-1 subspace. Instances and concepts are thus the same mathematical objects, differing only in dimension.
> - **Relation.** An instance $\mathbf{v}$ belongs to concept $\mathcal{S}$ if $\|\mathbf{P}_S\mathbf{v}\|\approx\|\mathbf{v}\|$. This is not mixing entities and propositions, rather it is a geometric interpretation of logic (Birkhoff & von Neumann, 1936), adapted for machine learning (van Rijsbergen, 2004). This is consistent with prior works that represent partial-orders via regions of an ambient space e.g., cone embeddings (Ganea et al., 2018), box embeddings (Vilnis et al., 2018), and Gaussian embeddings (Vilnis & McCallum, 2015). Instead of subspaces, these works parameterize regions with cones, boxes and Gaussian measures, respectively. This view of concepts as regions of space is described in the seminal work *Conceptual Spaces* (Gardenfors, 2000).
>
> ### Unifying Entities and Propositions
> The reviewer states *"It seems that authors mix entities and propositions, and this will introduce problems."*. We argue that this unification is a theoretical feature, not a flaw. Drawing from Quantum Logic (Birkhoff & von Neumann, 1936) and Distributional Semantics, we treat both entities (*e.g.*, "Man") and propositions (*e.g.*, "Man is fishing") as predicates over the ambient space. This is precisely the formulation of visual-semantic hierarchies in (Vendrov et al., 2016), where "Man" and "Man is fishing" are mapped to the same partially-ordered space. In our setup, this space is the set of subspaces of $\mathbb{R}^d$, with entailment modeled via subspace inclusion (van Rijsbergen, 2004):
> - $S_\text{man}\subset S_\text{living thing}$ (Entity hierarchy).
>
> - $S_\text{man is fishing}\subset S_\text{Man is doing an activity}$ (Propositional entailment).
>
> By using the same geometric object (subspaces) for both, our model can seamlessly handle logical composition across the spectrum of linguistic complexity, as evidenced by our performance on WordNet and SNLI.
>
> ### Clarification on Uniqueness and Subspace Capacity
> The reviewer expresses concern regarding uniqueness: *"In this way, the concept “man on a boat” is represented by the subspace span(x1, x2)... The “man” on a boat can also sit, stand, sleep, jump…"*. This comment highlights a **crucial misunderstanding** of how we use subspaces in our model. The reviewer seems to interpret the vectors $\mathbf{X}=[\mathbf{x}_1​,\mathbf{x}_2]$ as an enumeration of specific states. **This is not the case.**
> - **Continuous Nature.** A subspace is a continuous region of latent space, not a discrete bag of attributes. A rank-$k$ subspace (spanned by $\mathbf{X}$) contains an infinite number of vectors.
> - **Parameterization $\mathbf{X}$ vs Subspace.** The vectors in $\mathbf{X}$ are merely the parameters used to define the orientation and dimension of the concept. They do not enumerate the contents and are never used as embeddings. Crucially, Fig. 1 is just an illustrative example and we can revise it to ensure clarity.
> - **Infinite Variations.** The concept "Man on a boat" is a high-dimensional plane. The specific cases of a man on a boat *"sitting"*, *"standing"*, or *"jumping"* correspond to lower-rank subspaces. As long as these subspaces lie approximately within the "Man on a boat" subspace, the model correctly classifies them as entailing that concept. We do not need to add a new vector to $\mathbf{X}$ for every possible action. The subspace generalizes to cover the region of space where "man on a boat" is valid.
> - **Adaptive Dimensionality.** The model learns the rank needed during training. As shown in Fig. 5, generic concepts in WordNet are learned as high-rank subspaces (high capacity), while specific leaves are low-rank. The model automatically adjusts dimensionality to capture the "determinacy" the reviewer asks for.

---

> > ### Comment · Reviewer_Mkgu · 2025-11-27
> >
> > Thank you for the reply. Could you explain the following cases:
> >
> > What are the subspace of "a man fishing" and the subspace of "a woman fishing"? Do they intersect? What does this intersection mean?
> >
> > What are the subspaces of "Tom", "John", "Tom fishing", "The individual who is fishing is not Tom", "The individual who is fishing is John", "The present King of France is bald."?

---

> > > ### Author Response · Authors · 2025-11-27
> > >
> > > We thank the reviewer for raising fundamental questions regarding the logical soundness of our approach. We particularly appreciate the inquiry into Russell’s paradox, as it allows us to clarify an advantage of our subspace approach over standard vector embeddings.
> > >
> > > We emphasize that our method is a data-driven representation learning approach. The subspaces are the output of an encoder trained to minimize a loss function over training data, similar to Box embeddings (Vilnis et al., 2018), Gaussian embeddings (Vilnis & McCallum, 2015), Beta embeddings (Ren et al., 2020) and Quantum embeddings (Garg et al. (2019)). However, we can describe the **ideal geometry** that our method is capable of learning, which answers the reviewer's questions on disjointness and non-existence.
> > >
> > > Let us consider an NLI-trained model, akin to the one used in the paper.
> > >
> > > **Disjointness and Individuals (“Man” vs “Woman”, “Tom”, “John”).** We demonstrated empirically that **entailment** is represented as **subspace inclusion** and **contradiction** by **orthogonality** (see NIS Histograms in Fig. 3). Under SNLI labeling, *“a man fishing”* contradicts *“a woman fishing”*. Thus, **we should expect the corresponding subspace representations to be orthogonal and have a null intersection**. In general, we could expect the following structure to emerge:
> > >
> > > - $S_\\text{Tom fishing} \\subset S_\\text{Tom} \\subset S_\\text{man}$
> > > - $S_\\text{John fishing} \\subset S_\\text{John} \\subset S_\\text{man}$
> > > - $S_\\text{Tom fishing} \\subset S_\\text{a man fishing} \\subset S_\\text{a individual fishing}$
> > > - $S_\\text{John fishing} \\subset S_\\text{a man fishing} \\subset S_\\text{a individual fishing}$
> > >
> > >
> > > The subspace $S_\\text{an individual is fishing}$ **includes all entailing variants**: $S_\\text{a man is fishing}$, $S_\\text{John is fishing}$, $S_\\text{Tom is fishing}$, etc. From this region of space, we can remove *"Tom"*, to obtain $S_\\text{an individual is fishing but not Tom} = S_\text{an individual is fishing} \\cap S_\\text{Tom}^\perp$. This carves out *“Tom”* from the *"fishing"* subspace. Suppose the training data is such that all sentences entailing (and thus included in) *“a man fishing”* are in fact variants of *“Tom is fishing”*. In this case, the subspace of *“a man fishing”* would equal that of *“Tom fishing”*. Therefore, $S_\\text{an individual is fishing but not Tom}$ would be the null subspace.
> > >
> > > **Non-Existing References (“The King of France”).** We appreciate the reference to Bertrand Russell’s puzzling example *"The present King of France is bald."*. This question highlights a **structural advantage** of our subspace formalism over standard vector embeddings:
> > > - **Standard Embeddings**: In standard vector models, *"King of France"* must be mapped to a vector in space $v\\in\\mathbb{R}^d$. This vector necessarily has a magnitude and direction, incorrectly implying existence and semantic validity comparable to *"King of England."*.
> > > - **Subspace Embeddings (Ours)**: Our framework admits a representation for non-existent or empty concepts: the **Zero Subspace** ({0}).
> > > If the model correctly learns that there is nothing simultaneously entailing *“King”* and *“France”*, the intersection of *"King"* and *"France"* is empty ($S_\\text{King}\cap S_\\text{France}​=\\{0\\}$) *i.e.,* the resulting representation is the null subspace. Consequently, any further intersection applied to it (*e.g.*, *"is bald"*) projects onto the zero subspace too, implying non-existence according to the model.
> > >
> > > Thus, while we do not claim our SNLI-trained model knows French history, we claim that our formalism provides a rich geometric vocabulary, which allows us to represent non-existent referents, which standard vector geometry lacks.

---

> > > > ### Comment · Reviewer_Mkgu · 2025-11-27
> > > >
> > > > $S_\text{Tom}$ is open or close ?
> > > >
> > > > "The present King of France is bald." is not a paradox. Your answer seems that if your system is trained properly, it will return empty space.  The problem is that there were kings in France. Here, you need to represent temporal concepts. Are spatial and temporal concepts, such as yesterday, Mondays, Frances, Paris, Rome, all embedded as subspaces?  How to embed this Monday and on Mondays?
> > > >
> > > > If every concept is represented as subspaces, "Socrates is human" will be encoded as
> > > > $S_\text{Socrates}\subset S_\text{human}$.  How to distinguish membership relation from subset relation?

---

> > > > > ### Author Response · Authors · 2025-11-27
> > > > >
> > > > > We thank the reviewer for continuing the discussion. We emphasize that our work should be evaluated under the standard assumptions of **representation learning**, where **all concepts are learned from data through a shared parametric form** (vectors, Gaussians, cones, boxes, or in our case subspaces).
> > > > >
> > > > > ### “$S_\text{Tom}$ is open or close ?”
> > > > > In a finite-dimensional vector space like the ones we employ (subspaces of $\mathbb{R}^d$), **every linear subspace is a closed set**. Importantly, this property plays no role.
> > > > >
> > > > > ### On Temporal and Spatial Concepts
> > > > > As in the geometric embedding literature, the encoder assigns a geometric object to every input string. Thus, if the training data contains expressions such as *“yesterday”*, *“on Mondays”*, *“Paris”*, or *“Rome”*, the model outputs subspaces for those expressions in exactly the same way sentence encoders output vectors. **No hand-crafted temporal or spatial rules are required or assumed.** This is fully consistent with prior work, which also treats everything as learnable geometric objects (vectors, boxes, cones, Gaussians, points in Hyperbolic space, etc.) without hand-engineering time or space. Returning to the "King of France" example, if the training data contemplates such examples, $S_\text{Louis XIV}$ could very well be included in $S_\text{King of France}$ while having zero intersection with $S_\text{Present day king of France}$, which would be the zero subspace. While incorporating explicit temporal or spatial structure into geometric embeddings may be an interesting direction for future research, such extensions are beyond both the scope of this work and the assumptions of the existing literature we build upon.
> > > > >
> > > > > ### How “Socrates is human” is represented
> > > > > In our framework, the subspace *“Socrates”* would be included in the subspace *“human”*. The statement *“Socrates is human"* can be seen as the inclusion relationship, akin to how box and cone embeddings use box and cone inclusion and Gaussian embeddings use KL-based inclusion.
> > > > >
> > > > > ### Scope of the Paper
> > > > > We clarify that our contribution lies in representation learning, not symbolic knowledge engineering. Questions about encoding arbitrary temporal, spatial, or philosophical constructions assume a level of explicit, hand-specified semantics that all learned representation methods, including vectors, hyperbolic space, Gaussians, boxes, and cones, abstract away from. Our goal is to introduce a geometric inductive bias (subspaces) that supports inclusion and hierarchical structure and that achieves strong empirical performance on standard benchmarks.

---

> > > > > > ### Comment · Reviewer_Mkgu · 2025-11-28
> > > > > >
> > > > > > Thanks for the continued explanation. The work is in representation learning and support $\land$, $\lor$, and $\neg$ operations.
> > > > > >
> > > > > > If every linear subspace is a closed set, a subspace and its complement will have non-empty intersections. You cannot solve empty-reference problems through logical operation, such as "the present French king".  If the subspace is learned from data, it will not be empty space.
> > > > > >
> > > > > > Spatial and temporal concepts are not hand-crafted concepts. They are as normal as Tom, and Paris, $S_{Paris}\subset S_{France}$, $S_{every Monday}\subset S_{every\ day\ in\ a\ week}$.
> > > > > >
> > > > > > If the statement “Socrates is human" can be seen as the inclusion relationship, the membership relation will be mistakenly represented as the subset relation. As you mentioned Russell's paradox, your method cannot represent Russell's paradox correctly, again, either through subspace operations or through representation learning. $x\in x$ will be mistakenly represented as $S_x \subset S_x$.

---

> > > > > > > ### Author Response · Authors · 2025-12-02
> > > > > > >
> > > > > > > **We thank the reviewer again for their continued engagement. We address the remaining concerns while clarifying the scope of our contribution.**
> > > > > > >
> > > > > > > Regarding the concern that *“a subspace and its complement will have non-empty intersections. You cannot solve empty-reference problems through logical operation”*, we note that every subspace of $\\mathbb{R}^d$ contains the origin, so any two subspaces intersect at $\\{\\mathbf{0}\\}$. **This behavior is expected and does not conflict with the intended semantics.** In the orthomodular lattice of Hilbert-space subspaces, the zero subspace represents the **identically false** or **absurd** proposition, while the full space represents the **identically true** proposition (Prop. S4 in pp 828 from Birkhoff & von Neumann, 1936 and pp. 162–163 from von Neumann, 2018). **Subspaces whose only intersection is the origin correspond to contradictory or empty concepts.**
> > > > > > >
> > > > > > > In our setting, semantics depend on whether two subspaces share **non-zero vectors**, since these represent non-trivial instances. Hence, the expected correspondences hold:
> > > > > > > - Disjoint sets $A\\cap B=\\varnothing$ correspond to $\\mathcal{S}_A\\cap \\mathcal{S}_B=\\{\\mathbf{0}\\}$.
> > > > > > > - $A\cup \\varnothing =A$ corresponds to $\\mathcal{S}_A+\\{\\mathbf{0}\\}=\\mathcal{S}_A$​.
> > > > > > > - $A\cap \\varnothing =\\varnothing $ corresponds to $\\mathcal{S}_A\\cap\\{\\mathbf{0}\\}=\\{\\mathbf{0}\\}$​.
> > > > > > >
> > > > > > > This behavior also appears empirically. As shown in Fig. 4, subspace rank correlates strongly with concept generality, and **zero-rank subspaces behave as empty concepts, containing no non-trivial vectors**. In our WordNet embeddings, whenever two nodes $i, j$ satisfy $\\text{Hyponyms}(i)\\cap\\text{Hyponyms}(j)=\\varnothing $, we indeed observe $S_i \\cap S_j \\approx \\{\\mathbf{0}\\}$. Further, the subspace encoder we propose can output the zero subspace when appropriate.
> > > > > > >
> > > > > > > Finally, we emphasize that our method is **not intended as a symbolic logic engine**. Our setting follows the same assumptions as vector, hyperbolic, Gaussian, cone, and box embeddings, which all treat concepts as geometric objects learned from data. Within this established framework, our subspace representation subsumes vectors and provides **emergent compositionality** through subspace operations, a capability which is absent in Euclidean, hyperbolic, Gaussian, and box embeddings. This behavior is supported empirically: in our new composite entailment experiment, we achieve **over 90% AUC for both conjunctions and negations**, while all vector embeddings achieve **below 70% AUC on negations**.

---

> ### Author Response · Authors · 2025-11-23
>
> (part 2)
>
> ### Statistical Evaluation vs. Symbolic Rigour
> We argue that bridging the gap between rigorous logic and statistical learning is precisely our contribution. Purely symbolic systems are brittle and fail on noisy, real-world data (like SNLI), while standard neural networks have no logical structure. We impose a geometric structure (subspaces) that allows for logic (intersection, union, negation) while retaining the differentiability required for statistical performance. In fact, matching or exceeding SoTA on statistical benchmarks demonstrates that our subspace approach is structurally richer, but it is also competitive with, or superior to, vector embeddings. The emergent properties presented in Section 5 and evidenced by the new zero-shot compositionality experiment in Appendix E.1 cannot be achieved with vector embeddings.
>
> ### Topological Relations, Distances and Orientation
> - **Topology.** The reviewer suggests modeling relations like "on" topologically. While interesting, our focus is on hierarchical/logical relations (entailment, negation), which are foundational to reasoning. Crucially, we do not explicitly model any relation, we simply impose the subspace representation and the structure emerges from the data via standard loss functions (cross-entropy and InfoNCE). Spatial prepositions are a specific semantic domain while our framework is a general one.
> - **Distance and Orientation.** The reviewer asks about orientation. The **orientation and dimension of the subspace are the representation**. The distance between concepts is defined by the principal angles between their subspaces (which we use to quantify similarity and inclusion throughout the paper). **This is a direct generalization of the commonly used inner product (cosine) similarity.**

---

### Official Review · Reviewer_6uJV · 2025-11-02

**Soundness:** 3
**Presentation:** 4
**Contribution:** 4
**Rating:** 8
**Confidence:** 3

**Summary:**

This paper proposes representing linguistic concepts as learnable linear subspaces rather than fixed-dimensional vectors. Specifically, the authors argue that this paradigm enables one to naturally capture concept complexity through subspace dimensionality, inter-concept hierarchies through subspace inclusion, and multi-concept logical operations (conjunction, disjunction, negation) through simple algebraic manipulations. More importantly, this work introduces an efficient way of learning these subspaces in a differentiable manner. The proposed representation learning approach is evaluated, both quantitatively and qualitatively, on several NLP tasks. These experiments strongly suggest that the proposed paradigm outperforms competing state-of-the-art representation learning approaches while offering distinct new benefits such as naturally capturing inter-concept relationships and complexities.

**Strengths:**

Thank you so much for submitting this work! I enjoyed reading this paper and learned a great deal from it. Below are what I believe are this paper’s main strengths:

1. **[Originality, Critical]** The proposed paradigm (i.e., representing concepts as subspaces rather than fixed vectors), the used learning algorithm (i.e., a mechanism for learning different rank subspaces from the data), and the properties that derive from using this paradigm and learning process (i.e., naturally composable and hierarchical representations) are all, to the best of my knowledge, novel and interesting proposals. Because of this, I strongly believe the sheer originality of this paper is very strong.
2. **[Significance, Critical]** This work introduces an elegant geometric framework that naturally enables representations to capture hierarchies (inclusion), complexity (dimension), and logical compositions (i.e., via linear operations). These are all highly desirable properties that, as the authors correctly argue, are known to be problematic/lacking in existing representation learning frameworks. Therefore, I believe that the main ideas proposed in this paper have the potential to have a high impact across various areas in AI. In particular, I find the proposed differentiable mechanism for learning distinct ranks for different subspaces to be extremely interesting and potentially applicable to a wide range of representation learning tasks.
3. **[Quality, Major]**  The evaluation of the proposed method is extensive, spanning everything from traditional representation learning tasks to qualitative analyses and more interesting experiments that showcase novel features that emerge from the proposed paradigm. Moreover, the results suggest that the proposed methodology improves upon the SotA performance on key representation learning tasks (e.g., WORDNET reconstruction) while it provides a more interpretable, geometrically-grounded model of entailment. All of these are very strong forms of high-quality experiments that strongly suggest that the proposed methodology "works" as intended.
4. **[Clarity, Major]** The paper is very well-written and carefully motivates every key decision. Moreover, the supplementary material includes the code used for this paper, and the appendix contains concise and clear proofs of critical theoretical results/considerations that appear in this work.

**Weaknesses:**

In contrast, I believe the following are some of this work’s limitations:

1. **[Significance, Major]** Against standard good scientific practices, the empirical quantitative experiments in this work do not include any error bars. This makes it difficult for one to judge the significance of any observed differences and sets a negative precedent for not following good experimental conventions.
2. **[Quality and Clarity, Minor]** It is unclear how certain hyperparameters like $ \lambda $ are selected and what their impact is on the observed results.
3. **[Quality and Clarity, Minor]** Although there is a short discussion on the effectiveness of the proposed paradigm, it is unclear how this manifests itself in practice in terms of quantities that are practically important (e.g., memory consumption, wall-clock training times, etc.).
4. **[Quality, Minor]** There is no discussion of any limitations of the proposed framework anywhere in the paper.

**Questions:**

Balancing the contributions and strengths of this paper with respect to the weaknesses listed above, I am leaning towards accepting this work. This is because I believe this is a well-written piece of work that proposes a very interesting, elegant, and practical paradigm, whose impact could be significant. Nevertheless, it is worth mentioning that my main area of research is not specifically in language representation learning, so it is possible I might’ve missed some important distinctions/previous works that this paper failed to consider (hence my relatively low confidence). That being said, I have included some questions below that could help improve the quality of this manuscript. If these concerns are properly addressed, particularly my concerns regarding the lack of error bars in the quantitative experiments, I would be more than happy to consider updating my score.

1. **[Major]** Could you please provide error bars for the empirical quantitative results? Are the improvements in SE's results significant?
2. **[Major]** How was the hyperparameter $\lambda$ chosen? What effect does this hyperparameter have on the observed results and stability of the training procedure?
3. **[Minor]** How does the learning time and memory usage of SE/SPH compare to that of competing baselines (in practical terms, such as wall-clock times and peak memory consumption rather than asymptotically)?
4. **[Minor]** Could you please describe some key limitations of SE? If possible, I would strongly recommend adding these limitations to the paper.
5. **[Minor]** Why is $\mathbb{R}^{10}$ used for the Euclidean baseline for the Link Prediction experiments rather than $\mathbb{R}^{128}$ to match the maximum rank of SE? Is that a fair comparison?

### Minor Suggestions and Typos

1. **[Potential Typo, Nit]** There are missing parentheses in the citations in lines 76 and 376.

---

> ### Author Response · Authors · 2025-11-23
>
> We thank the reviewer for their thoughtful and detailed feedback. We are glad that the originality, significance, and clarity of our work were appreciated. Below, we address the questions raised, in detail.
>
> ### Error bars
> We agree that reporting statistical uncertainty is important. We have rerun all quantitative experiments with 5 independent random seeds and now report mean ± standard deviation for all key metrics. Tables 4 and 5 have been updated accordingly, and the improvements of SE over the baselines remain statistically significant.
>
> ### Effect of lambda
> The hyperparameter $\lambda$ controls the smoothness of the soft projections. As expected, with $\lambda=0$ (no smoothing i.e., using true orthogonal projection operators), training collapses. However, we observed stable optimization across a wide range of $\lambda > 0$. In practice, we selected $\lambda$ via grid search over $\{0.01,0.05,0.1,0.2\}$ using the validation set. We will include an ablation study on $\lambda$ to clarify its effect.
>
> ### Computational Efficiency and Practical Overheads
> We acknowledge the importance of reporting practical training costs. In response, we have added a comparison of wall-clock training time per epoch and peak GPU memory usage in Appendix G (Table 8). These measurements were obtained under the same training setup as our NLI experiments. For the larger backbone (mpnet-base-v2), our method introduces a modest overhead: approximately $1.3\times$ slower and $1.2\times$ higher peak memory than the baseline. We emphasize that this overhead is limited to training: once trained, the encoding overhead is minimal. Recall from Table 9 that the overhead introduced by the SPH is 0.12ms/query (batch-size of 128). In addition, the representation can be vectorized and used with standard Euclidean search methods, yielding fast retrieval, as shown in Table 6.
>
>
>
> **GPU Average Enconding Time (ms/query)** (see Table 9)
>
> | **Model**                                                                   | **Time (ms/query)** |
> |-------------------------------------------------------------------|------------------------------|
> | **mpnet-base-v2**                                                  |              0.56             |
> | **(Ours) mpnet-base-v2 + SPH (SE$^{128}$)**    |              0.68             |
>
>
> **SNLI Training Wall-Clock Time and Peak GPU Memory** (see Table 8)
>
> | **Model**                                                                           | **Time (s/epoch)** | **Peak GPU Memory (MB)** |
> |-------------------------------------------------------------------------------------------------------------------|----------|----------------|
> | **mpnet-base-v2 + MLP ($\mathbf{v_p}, \mathbf{v_h}, \mathbf{v_p}-\mathbf{v_h}$)**  |  551     | 32,328       |
> | **(Ours) mpnet-base-v2 + SPH (SE$^{128}$)**                                                              |   711    | 39,399       |
>
>
> ### Limitations
> We appreciate the suggestion to make the trade-offs explicit. We have added a "Limitations" section in Appendix H of the revised paper, highlighting two key areas:
>
> - **Approximate Logic.** Our use of soft projectors (with eigenvalues in $[0,1)$), as opposed to true orthogonal projectors (with binary eigenvalues), is essential for end-to-end differentiability. However, this implies that logical operations are approximate rather than symbolically exact, as evidenced by the error bounds in Table 1. A promising avenue for future research is to anneal the regularization parameter $\lambda \to 0$ during training. This would allow the model to leverage soft gradients for optimization while progressively converging toward exact orthogonal projectors, thereby enabling precise symbolic logic for downstream applications.
>
> - **Storage Complexity.** A possible concern regarding our soft projector representations is the memory footprint, if one were to store the full $d\times d$ matrices. However, as demonstrated in Fig. 5, the learned subspaces are highly compressible. We can explicitly enforce this low-rank structure during inference by storing only the top-k eigenvectors, thus bringing the average memory usage close to standard vector embeddings, while preserving downstream performance.
>
>
> ### Euclidean baseline
> We initially chose a 10-dimensional Euclidean baseline to match the dimensionality used in prior work. However, we agree that a higher-dimensional Euclidean baseline provides a more comprehensive comparison. We will update Table 4 to show that our method still outperforms a 128-dimensional Euclidean baseline.

---

> > ### Comment · Reviewer_6uJV · 2025-11-27
> >
> > Dear Authors,
> >
> > Thank you so much for taking the time to go over my review and for submitting this rebuttal. The description above and the paper edits satisfactorily addressed the questions I raised in my initial review. The only extra thing I would suggest is to please take advantage of the additional page you will get to include some of these edits (e.g., the limitations) in the main paper. Otherwise, having them in the appendix means they are likely to be entirely missed by readers (and, arguably, limitations should always be discussed in the main body of the paper).
> >
> > As for my score, I am still very much on the positive acceptance side of things. Currently, **my score could be said to be a "9", between poster and oral** (which unfortunately I can't indicate in the system used here). The only reason I don't dare move this entirely to an oral score (i.e., a 10) is that this isn't my area of expertise so that I might've missed something critical. Nevertheless, I have reviewed the other reviews of this paper, and my strong position has not changed as a result (in fact, I politely disagree with some of the assessments in the negative reviews after considering your rebuttals). As such, I look forward to discussing this paper with my fellow reviewers and wish you the best of luck.

---

> > > ### Author Response · Authors · 2025-12-02
> > >
> > > **We are pleased that the revisions have further strengthened the reviewer’s confidence in the paper.**
> > >
> > > Following your recommendations, we have **integrated all major updates directly into the main paper**, making use of the 10th page, rather than leaving them in the Appendix. In particular:
> > > - The new **Composite Entailment** experiment now appears in the main text (Section 4.4).
> > > - The **Limitations** section has also been added to the main body (Section 7), as suggested.
> > >
> > > We have also included a **study on the effect of the $\lambda$ hyperparameter** in Appendix E.1 (Table 8), showing that performance remains robust across a wide range of values ({0.01, 0.05, 0.1, 0.2}).
> > >
> > > **Note:** We have updated the global comment with the revised list of changes and the correct numbering of Sections and Tables.

---

### Author Response · Authors · 2025-11-23
**Overview of Paper Changes**

We thank the reviewers for their valuable feedback. We have uploaded a revised paper with the following additions:

- **Means and standard-deviations** (5 runs) added to Tables 4 and 5, as suggested by **6uJV**.

- **Limitations are described in Section 7 of the main paper**, as suggested by reviewers **6uJV** and **svE7**. We highlight two key areas for future improvements: 1) the approximate nature of the logical composition and 2) the storage complexity from naively storing $d\times d$ matrices.

- **Citations to prior works on Quantum Embeddings**, suggested by reviewer **kMKP**.

- **Training wall-clock time and peak GPU memory experiments in Appendix G (Table 9)**, as suggested by **6uJV**. For the larger models, the overhead is modest (1.3x slower and 1.2x more memory).

**Table: SNLI Training Wall-Clock Time and Peak GPU Memory** (see Table 9)

| **Model**                                                                           | **Time (s/epoch)** | **Peak GPU Memory (MB)** |
|--------------------------------------------------------------------------------------------------------------|---------------|----------------|
| **mpnet-base-v2 + MLP ($\mathbf{p}, \mathbf{h}, \mathbf{p}-\mathbf{h}$)**  | 551      | 32,328       |
| **(Ours) mpnet-base-v2 + SPH (SE$^{128}$)**                                                            |   711    | 39,399      |

- **Zero-shot experiment of composite entailment in Section 4.4 (Table 7)**. We quantify the emergent logical compositionality, as mentioned by **kMKP**. We compute 2-way entailment classification for composite hypotheses featuring conjunctions and negations, comparing against vector baselines. While all our embeddings retain the predictive performance of SNLI (90-97% AUC for both conjunctions and negations), vector embeddings experience considerable performance degradation (83-91% AUC for conjunctions and 49-69% AUC for negations).

**Table: Zero-Shot Composite Entailment AUC** (see Table 7)

| **Model**                                                                               | **AUC (Conjunctions)** | **AUC (Negations)** |
|-------------------------------------------------------------------------------------------------------------------|-------------|-------------|
| **mpnet-base-v2 + MLP ($\mathbf{p}, \mathbf{h}, \mathbf{p}-\mathbf{h}$)**  | 90.20      | 68.69      |
| **(Ours) mpnet-base-v2 + SPH (SE$^{128}$)**                                                              |   **96.55**    | **95.76**      |

- **Sensitivity to hyperparameter $\lambda$ in Appendix E.1 (Table 8)**, as suggested by **6uJV**. Across the tested range {0.01, 0.05, 0.1, 0.2} model accuracy varies only marginally.

**Table: SNLI test accuracy for all-miniLM-L6-v2 + SPH (SE$^{128}$) for different $\lambda$** (see Table 8)

|                       |0.01 |0.05 | 0.1 | 0.2 |
|-------------------|---------|-------------|-------------|-------------|
| **2-way**  | 91.18      | 91.26   | 91.12 | 91.06 |
| **3-way**  | 85.27 | 85.34 | 85.61 | 85.62 |

**Note:** Sections and Table numbers have been updated to reflect the numbering in the revised paper.

---

### Meta-Review · Area_Chair_u4LN · 2025-12-25

**Summary:**

The paper proposes a new approach which uses subspaces for constructing embeddings of concepts. It demonstrates good empirical results obtained with the approach across benchmarks.

The paper received divergent reviews with high variance in the scores. Reviewer 6uJV and svE7 (with scores of 8) liked the approach and the strength of the empirical findings. Reviewer Mkgu (score of 2) had fundamental disagreements about the proposed approach, and Reviewer kMKP (score of 4) felt that the approach lacked novelty, and that the empirical findings were lacking.

I believe there is substance behind each of the previous claims, both the positive and negative. The framework has some appeal compared to traditional embeddings, and is tested on several benchmarks. At the same time, the authors have not articulated as clear of a conceptual advantage over vector embeddings, as asked by Reviewer Mkgu. One could say this approach should only be validated by the results, but then it loses the appeal of having more rigor compared to other approaches (which is one of the claims of the paper). As Reviewer kMKP said, I think some comparison with LLM embeddings would be very valuable. LLMs are the default approach to learning embeddings, and some may argue that their embeddings already encode concepts such as inclusion and hierarchy via appropriate directions in the embedded space. Therefore comparing with LLMs would strengthen the paper.

Overall, the paper is borderline and can be accepted if there is room, but short of that I would recommend rejection.

**Reviewer Concerns:**

Concerns of Reviewers Mkgu and kMKP, discussed above, are mainly outstanding.

**Reviewer Scores:**

The scores would possibly have remained constant for this paper, based on the discussion and the concerns.

---

### Decision · Program_Chairs · 2026-01-26

Reject